# Early alterations of large-scale brain networks temporal dynamics in young children with autism

Aurélie Bochet [1,4✉], Holger Franz Sperdin [1,4✉], Tonia Anahi Rihs[2], Nada Kojovic[1], Martina Franchini[1], Reem Kais Jan[3], Christoph Martin Michel[2] & Marie Schaer[1]

Autism spectrum disorders (ASD) are associated with disruption of large-scale brain network. Recently, we found that directed functional connectivity alterations of social brain networks are a core component of atypical brain development at early developmental stages in ASD. Here, we investigated the spatio-temporal dynamics of whole-brain neuronal networks at a subsecond scale in 113 toddlers and preschoolers (66 with ASD) using an EEG microstate approach. We first determined the predominant microstates using established clustering methods. We identified five predominant microstate (labeled as microstate classes A–E) with significant differences in the temporal dynamics of microstate class B between the groups in terms of increased appearance and prolonged duration. Using Markov chains, we found differences in the dynamic syntax between several maps in toddlers and preschoolers with ASD compared to their TD peers. Finally, exploratory analysis of brain–behavioral relationships within the ASD group suggested that the temporal dynamics of some maps were related to conditions comorbid to ASD during early developmental stages.

[1] Department of Psychiatry, University of Geneva, Geneva, Switzerland. [2] Functional Brain Mapping Laboratory, Department of Fundamental Neurosciences, University of Geneva, Geneva, Switzerland. [3] College of Medicine, Mohammed Bin Rashid University of Medicine and Health Sciences, Dubai, United Arab Emirates. [4] These authors contributed equally: Aurélie Bochet, Holger Franz Sperdin. ✉email: aurelie.bochet@unige.ch; holger.sperdin@unige.ch

High-density electroencephalography (EEG) represents a powerful mean to explore the brain's physiological activity at a large-scale level in pediatric population[1]. Nevertheless, recording the brain activity during a task or at rest in very young children with autism spectrum disorders (ASD) is challenging. However, early identification of brain alterations is important as it provides insights onto the brain mechanisms that lead to their clinical behavioral phenotype. Ultimately, increasing our understanding on how differently the brain develops during childhood years can help clinicians to adapt and use more tailored therapies early in life when the brain is most plastic and thus responsive to behavioral treatment.

Recently, by combining high-density EEG and eye-tracking, we found that alterations in the directed functional connectivity between brain areas in the theta and alpha frequency bands are a core component of brain development at early stages of ASD[2]. Higher activity within key nodes of the *social brain*[3,4] for some toddlers and preschoolers with ASD was related to better visual exploration, and thus may represent a compensatory mechanism for ASD at such a young age[2,5]. Here, we used a data-driven, reference-free EEG microstate approach[6] to examine differences in the spatial organization and temporal dynamics of whole-brain neuronal networks in a large sample of toddlers and preschoolers with ASD and age-matched typically developing (TD) peers ($N =$ 113, 3 years of age on average). EEG microstates represent the sub-second coherent activation within global functional brain networks and are usually defined in the literature as short-lasting periods (approximately 100 ms) of quasi-stable topographies of the electric potentials in the ongoing EEG[7,8]. Interestingly, these rapidly changing EEG microstates are closely related and described as the electrophysiological correlates of functional MRI (fMRI) resting-state networks[9–11]. It is a commonly used method in the EEG field to study variations in the spatial organization and temporal dynamics of large-scale brain networks at rest or during a task. Microstate analysis has provided important insights on how differently the brain processes information in populations with brain disorders[6,12]. For example, numerous studies indicate that changes in the spatial and/or temporal characteristics of specific microstates represent critical markers for several brain disorders, indicating that these spatial and temporal modulations may mirror the information processing divergence in individuals with a neurodevelopmental condition compared to TD population[12–17].

Here, we hypothesized that the toddlers and preschoolers with ASD would show differences in the spatio-temporal properties of some microstates as well as the differences in transition probabilities compared to their TD peers. Considering the large heterogeneity in ASD in terms of autism symptom severity, developmental level and possible comorbid symptoms, we investigated brain–behavioral relationships between the temporal characteristics of the microstates and clinical phenotype. Finally, we used a bootstrapping approach[18] to examine the stability of our findings. Post hoc power analyses depending on the observed effect sizes were made to estimate the relationship between the sample size of our group and the observed statistical power. The bootstrapping procedure served to estimate the likelihood of finding the true result we observed in our full cohort from smaller sample sizes of participants.

## Results

**Microstate analysis**. The k-means cluster analysis across all participants identified five dominant maps, which explained 80.1% of the total variance (Fig. 1). The topographies of these five cluster maps corresponded to the canonical microstate classes previously reported in the literature and were labeled accordingly

(map A, B, C, D, and E)[6,19]. We also applied a k-means cluster analysis across toddlers and preschoolers with ASD and TD peers separately and extracted the five cluster maps for both groups. For both groups, the five cluster maps also corresponded to the canonical microstate classes previously reported in the literature and were therefore labeled accordingly (map A, B, C, D, and E). The five cluster maps extracted from ASD group explained 80.8% of the total variance of EEG recordings of this group, and the five cluster maps extracted from TD group explained 81.9% of the total variance of EEG recordings of this second group. The five cluster maps extracted from both groups separately were highly similar (Pearson's spatial correlations coefficients for map A = 0.992, map B = 0.986, map C = 0.996, map D = 0.958, and map E = 0.985) and were retained for further analysis (Fig. 2).

When looking for group differences in temporal parameters for the five maps, we found major differences in the temporal parameters of map B between toddlers and preschoolers with ASD and their TD peers (Fig. 3). All four parameters—global explained variance (GEV), mean duration, time coverage, and occurrence—of the map B were significantly increased in toddlers and preschoolers with ASD in comparison to their TD peers (GEV: $p < 0.001$; mean duration: $p < 0.001$; time coverage: $p < 0.001$; occurrence: $p < 0.001$). These differences survived false discovery rate (FDR) correction. As already mentioned, these differences in map B parameters also persisted when all females (11 ASD, 8 TD) were removed from the analysis (GEV: $p < 0.001$; mean duration: $p < 0.001$; time coverage: $p < 0.001$; occurrence: $p < 0.001$). We also found differences in temporal parameters of map E between ASD and TD groups for GEV ($p = 0.031$), time coverage ($p = 0.034$), and occurrence ($p = 0.019$) but not for the mean duration ($p = 0.162$). However, these differences were not significant after FDR correction. There were no significant differences in temporal parameters for map A, C, and D between both groups ($p$-values ranging from $p = 0.086$ to $p = 0.903$). There was no effect of age on the results, as correlations between temporal parameters and age of participants were not significant ($p$-values range from 0.096 to 0.966). See Table 1 for temporal parameters values for both groups.

**Correlations with clinical measures**. The results of Pearson's correlations and Spearman's rank correlations suggested specific microstate classes were related to specific symptoms in the toddlers and preschoolers with ASD group. Indeed, when looking at ASD symptoms' severity, we found a negative correlation between ADOS social affects severity score and the mean duration of the map E (uncorrected $p$-value = 0.030). This suggests that children with more social difficulties, have a shorter duration of the map E. Considering the developmental level of children with ASD, we found negative correlations between the fine motor domain of the MSEL and three temporal parameters of the map D—GEV (non-corrected $p$-value = 0.006), time coverage (non-corrected $p$-value = 0.007), and occurrence (non-corrected $p$-value = 0.005). This suggests that the lower the child's fine motor skills are, the less prevalent their map D. Finally, considering the comorbid symptoms of children with ASD, we mostly found positive correlations between children's affective problems reported in Child Behavior Checklist (CBCL) 1.5–5 and three temporal parameters of the map B—GEV (non-corrected $p$-value = 0.003), time coverage (non-corrected $p$-value = 0.008), and occurrence (non-corrected $p$-value = 0.048). This suggests that the more emotional difficulties the child has, the more prevalent map B is. We also found negative correlation between children's attention deficits and hyperactivity problems reported in CBCL 1.5–5 and the mean duration of the map C (non-corrected $p$-value = 0.020), suggesting that the more attention

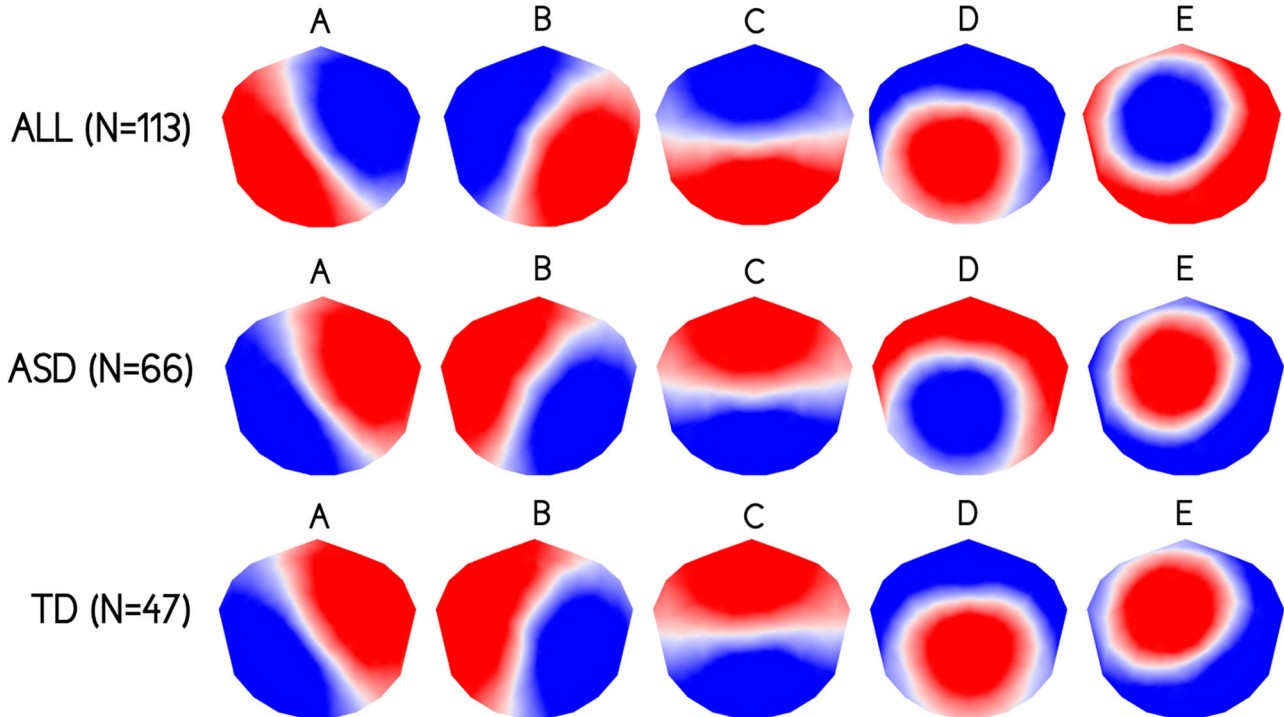

**Fig. 1 Microstate topographies.** The five microstate topographies identified in the global clustering across all subjects ($N = 113$), the autism spectrum disorders (ASD) group ($N = 66$), and the typically developing (TD) group ($N = 47$). Polarity of microstate topographies is not relevant meaning that topographies with opposite polarity are considered as the same microstate.

deficits and hyperactivity problems the child has, the shorter the duration of the map C is. However, correlations were not significant after FDR correction for multiple comparisons, and hence should be considered exploratory. Results of the exploratory correlations are reported in Fig. 4.

**Bootstrapping analysis**. The bootstrapping sub-sampling analysis tested the likelihood for observing a significant difference in the temporal parameters between both groups for each simulated sample size (Fig. 5). The results of stability analyses demonstrated a decrease in the likelihood of observing significant difference of temporal parameters between toddlers and preschoolers with ASD and TD peers as the sample size decreased. For instance, with a sample of 20 children with ASD and 20 TD children, a significant difference of Map B GEV was only detected in 50% of the simulated sub-samples. However, for the mean duration, the time coverage, and the frequency of occurrence, a 50% of likelihood for significant difference was reached with a sample of 12 children with ASD and 12 TD children.

**Transition probabilities**. Comparisons of all transition probabilities between each microstate map revealed increased transitions from microstate map C to map B in toddlers and preschoolers with ASD compared to their TD peers ($p < 0.001$) (Fig. 6). Transition probabilities were also increased in toddlers and preschoolers with ASD from microstate map E to map D ($p < 0.001$) and from map E to map B ($p = 0.0145$). In parallel, transition probabilities were increased in TD children compared to children with ASD from microstate map C to map D ($p = 0.0245$) and from map E to map C ($p < 0.001$). However, differences in transition probabilities from microstate map E to map B and from microstate map C to map D did not survive to FDR correction.

## Discussion

We applied a microstate analysis on EEG resting-state recordings acquired in toddlers and preschoolers with ASD and their TD peers ($N = 113$) and investigated modulations in four temporal parameters (the GEV, the mean duration, the frequency of occurrence, and the time coverage). The meta-criterion determined an optimal number of five template maps that best described the entire dataset explaining 81.0% of the global variance. The first four maps were visually identical in their spatial orientation to the canonical microstate classes A, B, C, and D previously reported in the literature[6,12]. The fifth map corresponds with microstate class E previously reported elsewhere[11,20]. We then identified the five prototypical maps that best described the dataset of toddlers and preschoolers with ASD and their TD peers separately and, as templates were highly similar across both groups, we compared temporal parameters using these five microstate classes. We found significant differences in the four temporal parameters of the microstate class B (the GEV, the mean duration, the frequency of occurrence, and the time coverage) between toddlers and preschoolers with ASD and their TD peers, map B is more prevalent in ASD group considering the four temporal parameters. We did not find any statistically significant difference regarding the four temporal parameters in other microstate classes (A, C, D, and E) but we found a tendency for map E to be less prevalent in toddlers and preschoolers with ASD regarding the GEV, the time coverage, and the occurrence.

Moreover, exploratory analysis of brain–behavioral relationships within the ASD group suggested that some maps are also related to specific autistic and comorbid symptoms expressed in toddlers and preschoolers with ASD. Indeed, we found associations between social autistic symptoms and microstate class E (mean duration), fine motor skills of the MSEL and microstate class D (GEV, time coverage, and occurrence), affective problems with microstate class B (GEV, time coverage, and occurrence), and finally attention deficits and hyperactivity problems with

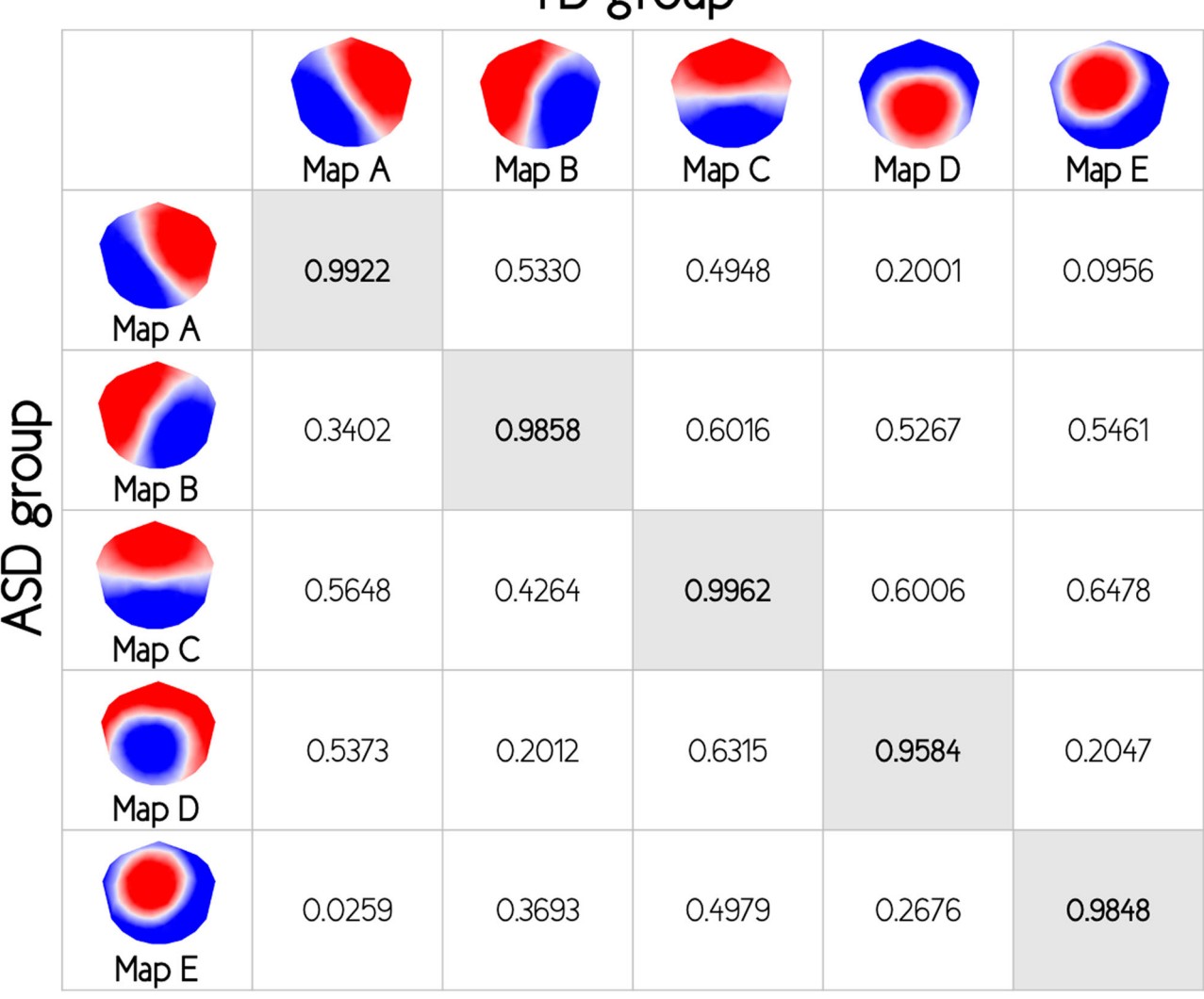

**Fig. 2 Spatial correlations between template maps.** Values of Pearson's spatial correlations coefficients between the 5 template maps of both autism spectrum disorders (ASD) and typically developing (TD) groups.

microstate class C (mean duration). However, none of the correlations were statistically significant after FDR correction for multiple comparisons.

Recently, we published an exploratory study that combined eye-tracking and microstate analysis in a small sample of children (N = 28)[19]. We found four group cluster maps, very similar to microstate classes A, B, C, and D previously described in the literature[19]. To the best of our knowledge, only six other studies using a microstate approach in individuals with ASD have been published[21–26]. Across resting-state studies, D'Croz-Baron and colleagues found six microstate classes that best described their dataset in young adults with ASD and their TD peers[21]. They found an increased occurrence for microstate classes B and E in individuals with ASD compared to their TD peers. There was a trend for microstate C being more present in their control group. However, this study included only 23 participants and only adults. Jia and Yu also highlighted a microstate class B more prevalent in occurrence and time coverage when comparing 5–18-year-old children with ASD to their TD peers[22]. Moreover, they found a decreased microstate class A regarding the mean duration and a decreased microstate class C regarding the mean duration and time coverage in ASD group. Finally, Nagabhushan Kalburgi and colleagues found an increased mean duration but a

decreased occurrence in microstate class C in children with ASD as compared to TD in an eyes-closed resting-state condition[26]. Both studies from Jia and Yu, and Nagabhushan Kalburgi et al. had small sample sizes (15 and 13 children with ASD, respectively) and a wide age ranges (5–18 y.o. and 8–14 y.o., respectively) in comparison to our study[22,26]. Our bootstrapping analysis demonstrated that a high level of likelihood (95%) to find a significant result for most of the temporal parameters of the map B was reached with samples above approximately 25 participants per group. This suggests that our sample size was sufficient in order to highlight significant differences between our groups of toddlers and preschoolers (N = 113).

In our study, microstate class B was more prevalent in the toddlers and preschoolers with ASD compared to their TD peers. The seminal EEG-MRI study of Britz and colleagues linked microstate class B with negative BOLD activation in bilateral extrastriate visual areas[9]. Exploring the functional significance of microstate classes, Milz and colleagues showed association of the microstate class B with verbal processing[27]. Resting-state brain networks (RSNs) are usually studied using fMRI because of its high spatial resolution[28]. Guo and colleagues used fMRI technique to study the brain dynamic connectivity and brain–behaviors relationships in ASD and they found

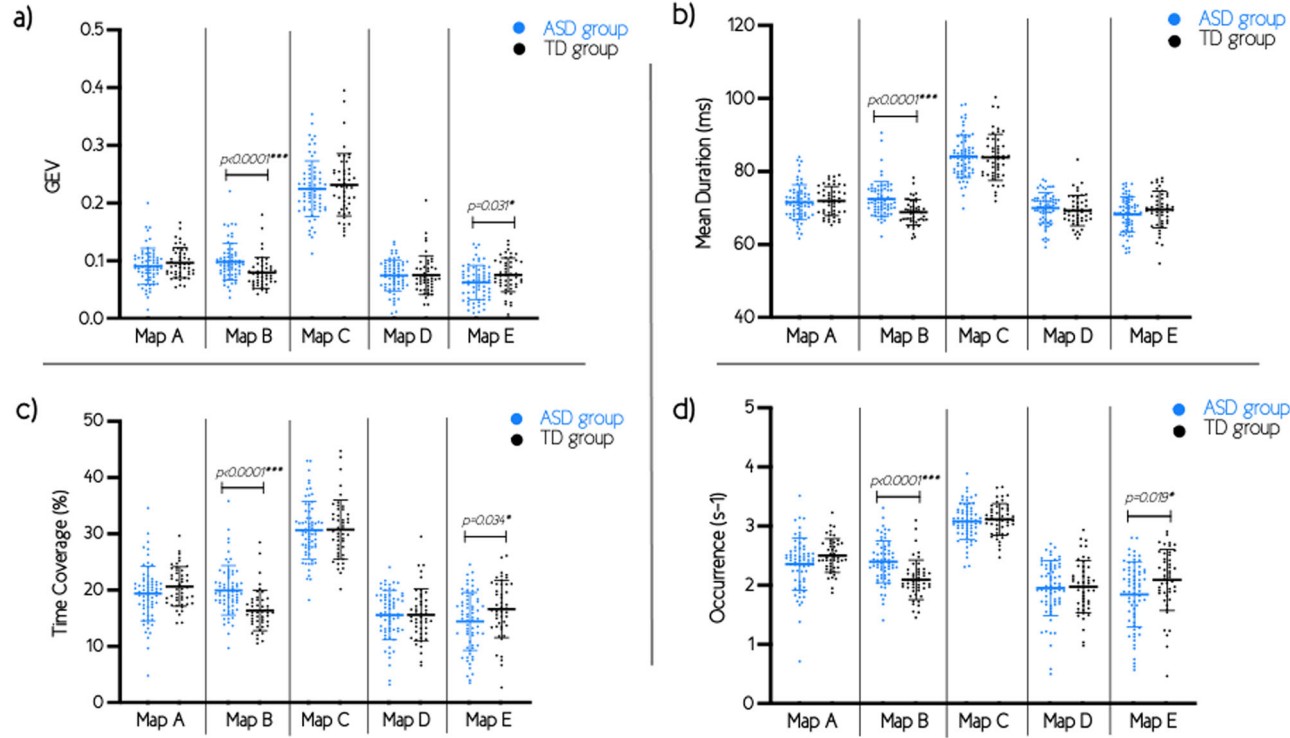

**Fig. 3 Temporal parameters of the microstates.** Results of both autism spectrum disorders (ASD) and typically developing (TD) groups for the temporal parameters: **a** global explained variance (GEV); **b** mean duration; **c** time coverage; and **d** occurrence. Error bars represent means and standard deviations. Uncorrected *p*-values. ***p*-values that survive false discovery rate (FDR) correction for multiple comparisons.

| **Table 1 Temporal parameters for the five maps of autism spectrum disorders (ASD) group and typically developing (TD) group.** | | |
|---|---|---|
| **Parameters** | **Autism spectrum disorders** | **Typically developing** |
| | **Mean, SD, N** | **Mean, SD, N** | ***p*-Value** |
| Map A - Global explained variance | 0.0909, 0.0310, 66 | 0.0967, 0.0261, 47 | = 0.353 |
| Map A - Mean duration (ms) | 71.58, 4.804, 66 | 71.91, 3.985, 47 | = 0.520 |
| Map A - Time coverage (percentage) | 19.40, 4.853, 66 | 20.65, 3.508, 47 | = 0.138 |
| Map A - Occurrence (s-1) | 2.357, 0.4418, 66 | 2.503, 0.2859, 47 | = 0.086 |
| Map B - Global explained variance | 0.0983, 0.0317, 66 | 0.0796, 0.0269, 47 | **<0.001*** |
| Map B - Mean duration (ms) | 72.48, 4.748, 66 | 68.86, 3.500, 47 | **<0.001*** |
| Map B - Time coverage (percentage) | 19.96, 4.390, 66 | 16.36, 3.584, 47 | **<0.001*** |
| Map B - Occurrence (s-1) | 2.398, 0.3540, 66 | 2.092, 0.3333, 47 | **<0.001*** |
| Map C - Global explained variance | 0.2249, 0.0483, 66 | 0.2317, 0.0540, 47 | = 0.560 |
| Map C - Mean duration (ms) | 84.12, 5.810, 66 | 83.87, 6.313, 47 | = 0.826 |
| Map C - Time coverage (percentage) | 30.64, 5.100, 66 | 30.76, 5.221, 47 | = 0.903 |
| Map C - Occurrence (s-1) | 3.081, 0.3300, 66 | 3.111, 0.2675, 47 | = 0.589 |
| Map D - Global explained variance | 0.0747, 0.0270, 66 | 0.0751, 0.0336, 47 | = 0.438 |
| Map D - Mean duration (ms) | 70.01, 4.171, 66 | 69.30, 4.183, 47 | = 0.145 |
| Map D - Time coverage (percentage) | 15.57, 4.353, 66 | 15.61, 4.593, 47 | = 0.533 |
| Map D - Occurrence (s-1) | 1.951, 0.4640, 66 | 1.976, 0.4438, 47 | = 0.870 |
| Map E - Global explained variance | 0.0627, 0.0295, 66 | 0.0756, 0.0294, 47 | = **0.031** |
| Map D - Mean duration (ms) | 68.27, 4.808, 66 | 69.57, 5.057, 47 | = 0.162 |
| Map D - Time coverage (percentage) | 14.42, 5.152, 66 | 16.62, 5.057, 47 | = **0.034** |
| Map D - Occurrence (s-1) | 1.846, 0.5499, 66 | 2.090, 0.5178, 47 | = **0.019** |
| In bold, significant *p*-values < 0.05. *p*-values that survive false discovery rate (FDR) correction for multiple comparisons. | | |

partially impaired functional connectivity between right anterior insula and default mode network (DMN)[29]. However, these RSNs are sensitive to dynamic fluctuation[30] that are difficult to capture with fMRI because of its moderate temporal resolution (in the order of seconds) and the delayed hemodynamic response. As such, EEG is a valuable alternative technique to study RSNs owing to its sub-second scale temporal resolution[6,31]. The increased prevalence of microstate class B in toddlers and preschoolers with ASD could reflect the atypical inter-network connectivity between visual network (VIS) and other major RSNs amongst individuals with ASD reported in the literature. For example, Gao and colleagues suggested hyper-connectivity between VIS and DMN probably impacting on language abilities in ASD[32]. Morgan and colleagues also highlighted hyper-connectivity between VIS, DMN, and language network[33].

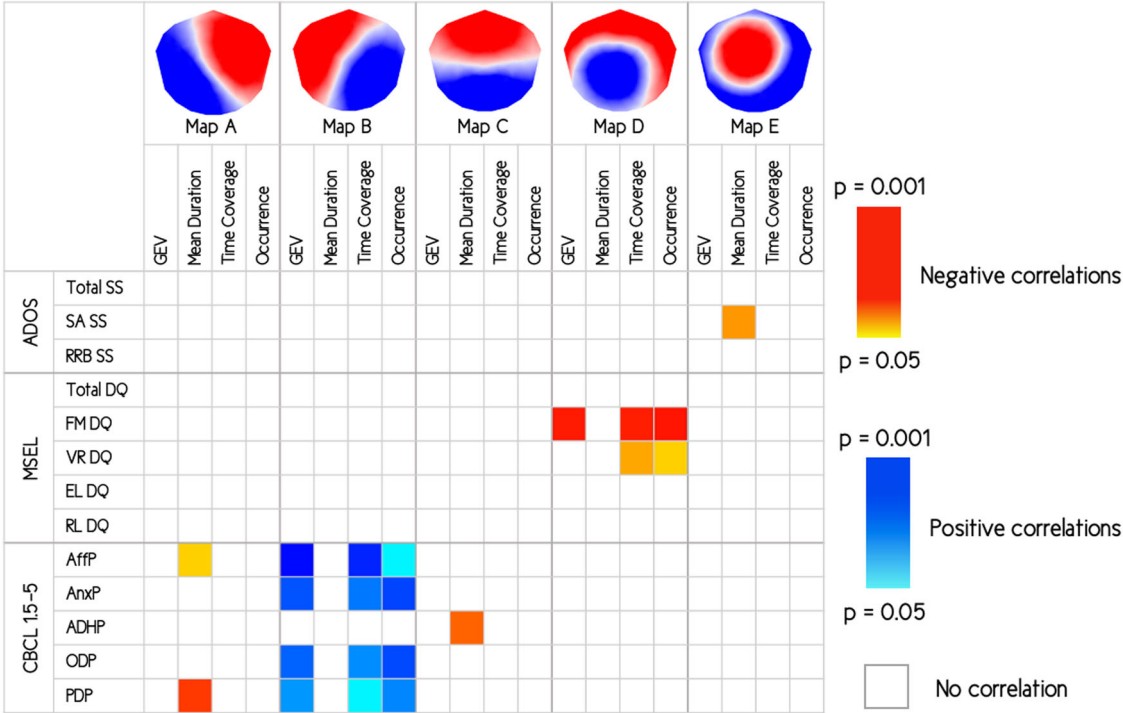

**Fig. 4 Correlations with clinical measures.** Matrix of exploratory correlations with clinical measures within autism spectrum disorders (ASD) group. Uncorrected *p*-values. ADOS = Autism Diagnostic Observation Schedule; SS = severity score; SA = social affect; RRB = repetitive and restricted behaviors; MSEL = Mullen Scales of Early Learning; DQ = developmental quotient; FM = fine motor; VR = visual reception; EL = expressive language; RL = receptive language; CBCL = Child Behavior Checklist; AffP = affective problems; AnxP = anxiety problems; ADHP = attention deficit/hyperactivity problems; ODP = oppositional defiant problems; PDP = pervasive developmental problems.

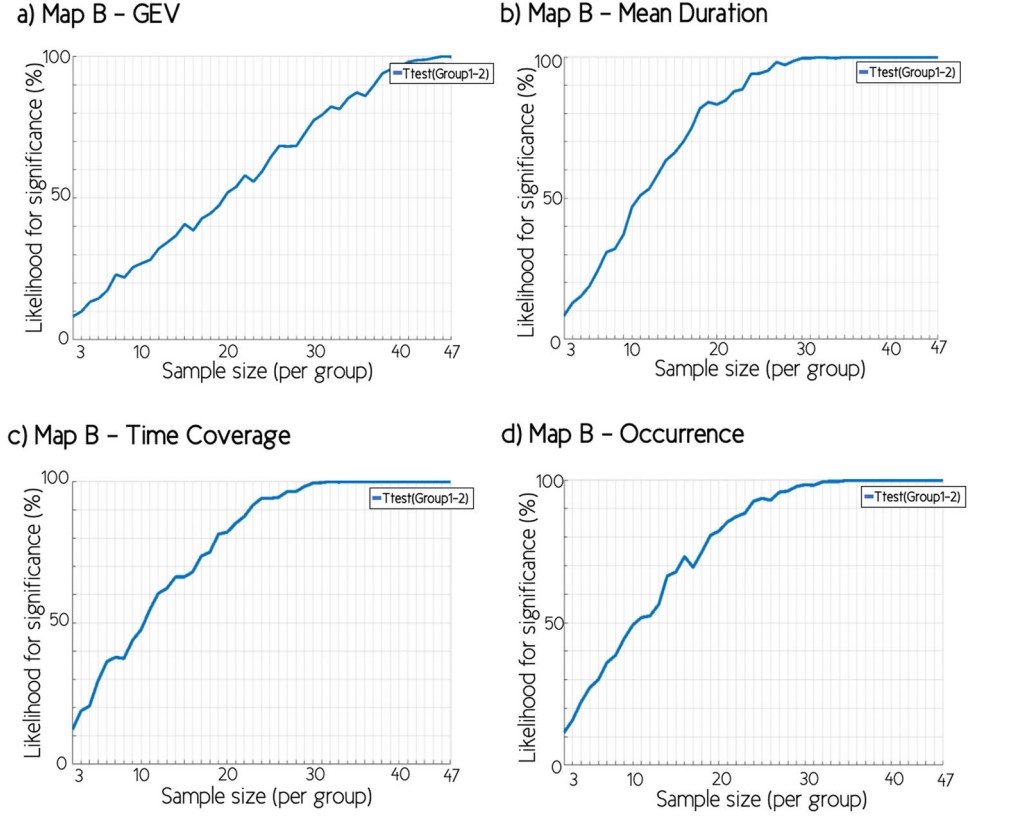

**Fig. 5 Bootstrapping analyses.** The likelihood to observe a significant difference between toddlers and preschoolers with autism spectrum disorders (ASD) and their typically developing (TD) peers, simulating sample sizes ranging from 3 to 47 individuals in each group, for parameters of microstate class B: **a** global explained variance (GEV); **b** mean duration; **c** time coverage; and **d** occurrence.

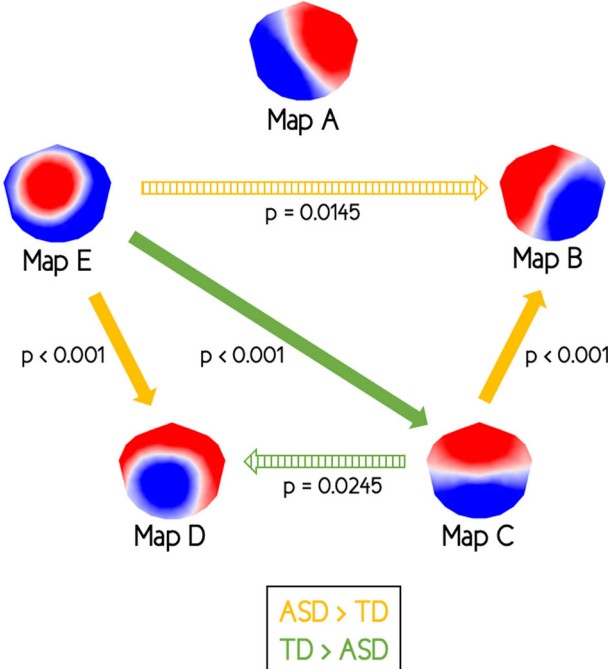

**Fig. 6 Dynamic syntax results using Markov chains.** Comparison of the transition probabilities between autism spectrum disorders (ASD) group and typically developing (TD) group. Uncorrected *p*-values. Orange arrows: transition probabilities in ASD group > TD group; green arrows: transition probabilities in TD group > ASD group; full arrows: *p*-values that survive false discovery rate (FDR) correction for multiple comparisons; hatched arrows: *p*-values that do not survive to FDR correction.

Moreover, transition probabilities normalized by occurrence of microstates revealed increased transitions from microstate map C to map B in toddlers and preschoolers with ASD compared to their TD peers, both maps reflecting activity in DMN and VIS, respectively, in the literature[9]. This result supports the hypothesis of hyper-connectivity between VIS and DMN in ASD[32,33]. We also found increased transitions from microstate map E to map D in toddlers and preschoolers with ASD compared to their TD peers. Conversely, transitions from microstate map E to map C were increased in TD group compared to ASD group.

Our result suggests that the temporal properties and the dynamic syntax of some specific microstate classes are already altered in ASD at early developmental stages. Highlighting brain differences early in life is important as it may ultimately help us to understand more what causes ASD and how the symptoms evolve over time given the vast heterogeneity of the ASD phenotype. We have started to characterize behavioral phenotypes in young children with ASD by taking developmental changes into account[34]. Currently, we are exploring how these brain-network differences evolve over the course of development as we want to find out how the brains of very young children with ASD can compensate and how these mechanisms emerge. This will ultimately lead to the development of more individualized and thus adapted therapies early in life when the brain is most plastic.

## Methods

**Participants**. This study was approved by the Local Research Committee, the Commission Centrale d'Ethique de Recherche (CCER) in Geneva, Switzerland, and written informed consent was obtained from all children's parents prior to inclusion in the study. In total, 293 participants were recruited for the experiment. We did not manage to place the cap on 108 participants. The cap was placed on the heads of 185 participants (110 ASD and 75 TD). Out of those, 60 participants (44 ASD and 16 TD) were excluded due to having too many movement-related artifacts, noisy signal, lack of interest, or having insufficient number of epochs for

subsequent analysis. This was to be expected given the sensory processing issues frequently reported in children with ASD (Kojovic)[34]. In order to minimize gender bias and include an equivalent percentage (20%) of females in both groups, we excluded 12 females from the TD group. As a result, the final sample consisted of 113 participants: 66 toddlers and preschoolers with ASD (11 females; mean age $3.3 \pm 1.0$ years, range 1.8–5.9) and 47 TD peers (8 females; mean age $3.3 \pm 1.2$ years, range 1.8–5.8). Groups did not differ by age ($p = 0.8838$) or sex ($p > 0.9999$). Five minutes of spontaneous EEG recordings were acquired for all the participants included in the study. All participants were recruited as a part of the Geneva Autism Cohort, a longitudinal cohort of young children[35,36]. Toddlers and preschoolers were included in the ASD group if the previously established clinical diagnosis was confirmed by exceeding the threshold limit for ASD on ADOS-G (Autism Diagnostic Observation Schedule-Generic)[37] or ADOS-2 (second version)[38]. The ADOS assessments were performed and scored by experienced clinicians working in the research team and specialized in ASD identification. For toddlers and preschoolers who were administered the ADOS-G assessment, the scores were recoded according to the revised ADOS algorithm[39,40] to ensure comparability with ADOS-2. The mean severity score at ADOS for the toddlers and preschoolers with ASD group was $7.67 \pm 1.83$. For the control group, TD toddlers and preschoolers were recruited through announcements in the Geneva community. They were also assessed by ADOS-G or ADOS-2 to ensure the absence of ASD symptoms, which would be an exclusion criterion. All TD participants had a minimal severity score of 1. Children were excluded from the control group if they presented any neurological/psychiatric conditions and learning disabilities according to parents' interview and questionnaire, or if they had a sibling or first-degree parent diagnosed with ASD. Considering the young age of participants, cognitive functioning was assessed using one of two standardized tools: the Mullen Scales of Early Learning composite score (MSEL)[41] or the Psycho-Educational Profile, third edition (PEP-3)[42]. The MSEL is a standardized assessment used to measure cognitive functioning for children from birth through age 68 months. Information about cognitive functioning includes five subdomains: visual reception, fine and gross motor skills, receptive and expressive language. The total development quotient (total DQ) is referred to as an estimate of overall intelligence. In comparison, the PEP-3 is a standardized tool to assess developmental level of children with developmental disorder, in particular ASD, between 2 and 7.5 year olds. It includes the same five domains of the MSEL but combines nonverbal and verbal intelligence scores into the same verbal and preverbal cognition scale and adds an imitation skills index. We used either the MSEL or the PEP-3, depending when the children were included in our protocol. The MSEL was added to our standardized battery for cognitive assessment only after 2015. As a result, in the current study, cognitive skills for 55 children with ASD were estimated using the MSEL, and for 10 children with ASD using the verbal/preverbal cognition scale of the PEP-3. Data on cognitive skills were missing for one child with ASD. Finally, to investigate other symptoms that could be present in children with ASD, all parents have completed the CBCL for ages 1.5–5 years' version[43]. The CBCL 1.5–5 consists of 100 questions exploring five DSM-scales as attention deficit/hyperactivity problems, anxiety problems, affective problems, oppositional defiant problems, and pervasive developmental problems.

See Table 2 for characteristics of study participants.

**Procedure and task**. The experiment was conducted in a quiet room. To help the children and their relatives to get familiar with the protocol, they received 2 weeks prior to their visit a kit containing a custom handmade EEG cap, pictures and a video illustrating the experiment. Participants were seated alone on a comfortable seat or on their parents lap in order to reassure them and keep them as calm as possible to avoid hand and body movements. Once seated, the experimenter measured the circumference of the head. The cap of the corresponding size was then prepared and gently placed on the participant's head. A couple of minutes were taken in order to allow the participants to settle into the experiment's environment and get used to the cap before starting the experiment. To best capture the child's attention during the experiment, we showed them an age-appropriate animated cartoon of their choice. The EEG expert ensured that the cap was accurately placed and electrodes adjusted to keep impedance values below 50 kΩ prior to starting the recording.

**EEG acquisition and preprocessing**. The EEG was acquired with a Hydrocel Geodesic Sensor Net (HCGSN, Electrical Geodesics, USA) with 129 scalp electrodes at a sampling frequency of 1000 Hz. On-line recording was band-pass filtered at 0–100 Hz using the vertex as reference. Data preprocessing and microstate analysis were done using Cartool (http://sites.google.com/site/cartoolcommunity/) and Matlab (Natick, MA). First, we down-sampled the montage to a 110-channel electrode array to exclude electrodes on the cheek and the neck since those are often contaminated with muscular artifacts. Data were filtered between 1 and 40 Hz (using Butterworth filters) and a 50 Hz notch filter was applied. Each file was then visually inspected to detect periods of movement artifacts. These periods were excluded. We performed Independent Component Analysis (ICA) on the data to identify and remove the components related to eye movement artifacts (eye blinks, saccades)[44,45]. Channels with substantial noise were interpolated using spherical spline interpolation for each recording. The cleaned data were down-sampled to 125 Hz, recalculated against the average reference and a spatial filter was applied.

**Table 2 Characteristics of study participants.**

| Characteristic | Autism spectrum disorders | Typically developing | |
| --- | --- | --- | --- |
| N | 66 | 47 | |
| | (M/F) | (M/F) | p-Value |
| Gender ratio | 55/11 | 39/8 | > 0.9999 |
| | **Mean, SD, N** | **Mean, SD, N** | |
| Age in years | 3.3, 1.0, 66 | 3.3, 1.2, 55 | = 0.8838 |
| ADOS Total SS | 7.7, 1.8, 66 | 1.0, 0.0, 55 | <0.0001 |
| ADOS SA SS | 6.7, 2.1, 66 | 1.0, 0.0, 55 | <0.0001 |
| ADOS RRB SS | 8.7, 1.8, 66 | 2.2, 1.9, 55 | <0.0001 |
| MSEL Total DQ | 73.4, 24.5, 55 | 110.4, 13.7, 35 | <0.0001 |
| MSEL FM DQ | 79.7, 21.3, 55 | 104.0, 12.6, 35 | <0.0001 |
| MSEL VR DQ | 86.4, 26.1, 55 | 122.7, 23.8, 35 | <0.0001 |
| MSEL EL DQ | 60.4, 28.5, 55 | 104.2, 21.2, 35 | <0.0001 |
| MSEL RL DQ | 67.0, 31.7, 55 | 113.8, 16.5, 35 | <0.0001 |
| CBCL AffP T-score | 58.7, 8.1, 49 | 53.0, 5.4, 34 | <0.0001 |
| CBCL AnxP T-score | 56.9, 9.7, 49 | 52.4, 4.9, 34 | = 0.0025 |
| CBCL ADHP T-score | 55.9, 5.9, 49 | 51.9, 3.3, 34 | = 0.0005 |
| CBCL ODP T-score | 54.9, 6.9, 49 | 53.7, 7.1, 34 | = 0.0969 |
| CBCL PDP T-score | 68.7, 12.0, 49 | 51.4, 4.1, 34 | = 0.0969 |

*ADOS* Autism Diagnostic Observation Schedule, *SS* severity score, *SA* social affect, *RRB* repetitive and restricted behaviors, *MSEL* Mullen Scales of Early Learning, *DQ* developmental quotient, *FM* fine motor, *VR* visual reception, *EL* expressive language, *RL* receptive language, *CBCL* Child Behavior Checklist, *AffP* affective problems, *AnxP* anxiety problems, *ADHP* attention deficit/hyperactivity problems, *ODP* oppositional defiant problems, *PDP* pervasive developmental problems.

Finally, EEG experts (H.F.S. and A.B.) reinspected all data to ensure that no artifacts had been missed.

**Microstate analysis.** The pipeline for the analysis is illustrated in (Fig. 7). We applied a k-means cluster analysis to the data of each subject to estimate the optimal set of topographies explaining the EEG signal. The clustering was applied only at local maxima of the global field power (GFP), which is calculated as the standard deviation of all electrodes at a given time point and represents time points of highest signal-to-noise ratio (see[46] for formulas). The polarity of the maps was ignored during the procedure. The k-means cluster analysis was first computed at the individual level and then across all participants (children with ASD and TD children together) to obtain the group cluster maps. In order to determine the optimal number of maps at both levels (both within and across subjects), we applied a meta-criterion that includes seven independent criteria. For a detailed description of these criteria, see[20] and[11]. Then we identified the five prototypical maps that best described the dataset of toddlers and preschoolers with ASD and TD peers separately. We computed Pearson's spatial correlation for each of the five microstate classes to ensure that those five microstate classes were similar across both groups. High spatial correlation coefficients indicated that the microstate classes were similar amongst both groups and the five cluster maps extracted from both groups separately were then fitted back to the original EEG of each subject, depending on their group. This way, spatial correlation was calculated between the cluster maps and each individual data point and that data point was labeled with the cluster map that showed the highest correlation. Polarity of the maps was again ignored for the back-fitting procedure. Data points that did not correlate more than 50% with a given group cluster map were marked as unlabeled. Only labeled data points were included in the analysis, meaning that unlabeled data points and periods marked as artifacts during the preprocessing were excluded.

Four temporal parameters of the microstates were computed for each individual recording: GEV, the mean duration, the time coverage, and the frequency of occurrence. The GEV is an estimate between [0; 1] of the explained variance of a given map, weighted by the GFP. The mean duration is the average duration in milliseconds that a given cluster map is continuously present. The time coverage is the percentage of total time for a given cluster map in the individual EEG recording. The frequency of occurrence represents the number of times per second that a given cluster map appears in the individual EEG recording[6].

**Statistics and reproducibility.** All data visualization and statistical analyses were carried out using IBM SPSS Statistics, Version 24.0 (IBM Corp., Armonk, NY, USA), GraphPad Prism, Version 9 (GraphPad Software, La Jolla, CA, USA, www.graphpad.com) and Cartool (http://sites.google.com/site/cartoolcommunity/).

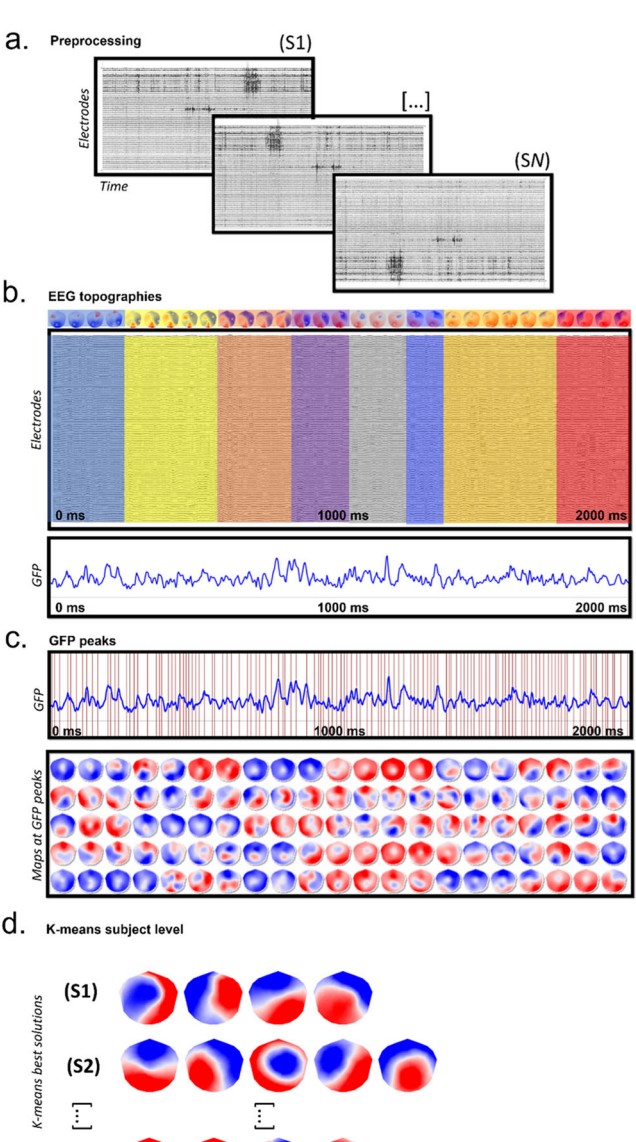

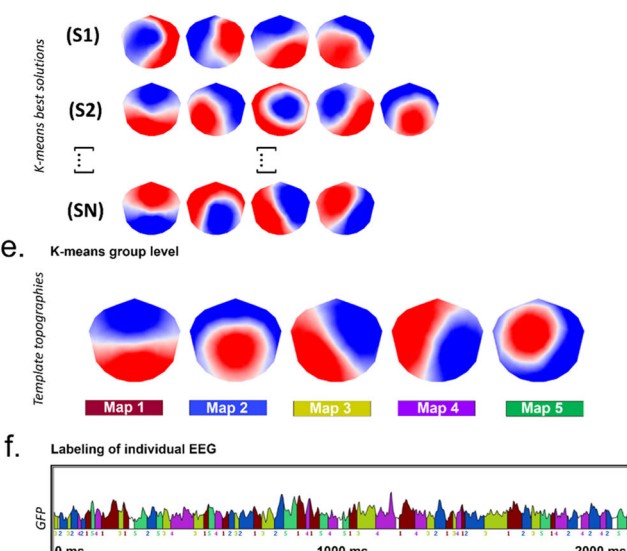

We first checked if the temporal parameters of the microstate classes had a normal distribution using D'Agostino and Pearson's tests. We investigated group differences using unpaired *t*-tests or Mann-Whitney-Wilcoxon pairwise tests if data were not normally distributed for each temporal parameter of each map. We applied FDR correction for multiple comparisons taking into account the four parameters of the five maps. We then conducted a male-only analysis (removed 19 female subjects; 11 ASD, 8 TD) as a validation of our attempt to control for gender bias. The findings from the male-only analysis were very similar to the larger analysis, indicating that the larger analysis is not biased by gender. To ensure that age had no effect on the results, we correlated the age of participants with temporal parameters, applying two-tailed Pearson's correlation or two-tailed Spearman's rank correlation, depending on the normality of distribution of the data.

**Fig. 7 Microstate analysis pipeline. a** Standard preprocessing of all the acquired high-density electroencephalography (EEG) recordings (110 channels). **b** A 2 s cleaned EEG and its corresponding global field power (GFP). Periods of quasi-stable map topographies (on top) are superimposed on the cleaned EEG and marked in different colors. **c** For each individual recording, peaks of GFP were determined (red vertical lines) and their specific potential maps were selected and submitted to a k-means clustering procedure (**d**). The best k-means clustering solutions at the individual level were selected based on the meta-criterion. **e** The best solutions obtained for each subject in step (**d**) were submitted altogether to a second k-means group cluster analysis. The meta-criterion identified a best solution with 5 template topographies (microstate classes). **f** The template topographies obtained in (**e**) were fitted back to the individual EEG recordings and each time point was labeled with the cluster map having the highest spatial correlation (winner-takes-all). The microstate sequence was used, for each subject, to extract the temporal parameters and statistical analysis.

Considering the large heterogeneity in ASD, we investigated possible brain–behavioral relationships between the temporal parameters of the microstate classes with children phenotype. Among toddlers and preschoolers with ASD, we correlated autism symptom severity, developmental level and comorbid symptoms with all temporal parameters. We applied two-tailed Pearson's correlation or two-tailed Spearman's rank correlation depending on the normality of distribution of the data between the GEV, mean duration, time coverage, and occurrence of microstates A, B, C, D, and E and Autism Diagnostic Observation Schedule total, social affect and repetitive and restricted interest severity scores (ADOS-G and ADOS-2)[37,38]. To explore the relationship between temporal parameters and the developmental level of children, we also applied correlation with the Mullen Scales of Early Learning composite score (MSEL)[41], using the total developmental quotient and fine motor skills, receptive and expressive language, and visual reception subdomains scales. We also explored comorbid symptoms by correlating temporal parameters with the five DSM-scales of the CBCL for ages 1.5–5 years' version parents' questionnaire[43]. The five DSM-scales are attention deficit/hyperactivity problems, anxiety problems, affective problems, oppositional defiant problems, and pervasive developmental problems. We applied FDR correction to correct for multiple correlations taking into account the four parameters of the five maps and the different clinical evaluations we used (ADOS, MSEL, and CBCL).

Then, always considering the large heterogeneity in the ASD phenotype, we wanted to estimate the likelihood of finding the significant results we observed in our full cohort from smaller sample sizes of participants. We therefore focused on microstate classes B. To do so, we simulated sample sizes ranging from 3 to 47 individuals in each group (with steps of 1 participant), using 500 bootstrapped sub-samples for each sample size. With each sample, unpaired $t$-test were performed to assess the significance of difference between both groups, using a statistical threshold of $p < 0.05$.

Finally, we analyzed the transition dynamics of EEG microstate maps by computing probabilities of all transitions from each microstate to any other using Markov chains. The observed probabilities were divided by the expected probabilities to account for the variability in occurrence of the states as described in the literature[11,47,48]. The transition probabilities between microstate maps were compared between ASD and TD groups applying unpaired $t$-tests and FDR correction.

**Reporting summary**. Further information on research design is available in the Nature Research Reporting Summary linked to this article.

## Data availability
The data that support the findings of this study are available on reasonable request from the corresponding author.

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

## Acknowledgements

The authors would like to thank all the families who took part in this research. This research was supported by the National Center for Competence in Research "Synapsy", financed by the Swiss National Science Foundation (SNF, #51NF40-185897) and by private funding by the Fondation Pole Autisme (http://www.pole-autisme.ch). This work was further supported by individual SNF grants to Marie Schaer (#163859 and #190084) and to Aurélie Bochet (#323530-183979), and a Marie Curie fellowship to Reem Jan, which received funding from the European Union Seventh Framework Program (FP7/ 2007- 2013) under grant agreement #267171.

## Author contributions

Conception and design of the experiment: H.F.S., T.A.R., and M.S.; Acquisition and processing of data H.F.S., A.B., T.A.R., R.K.J., M.F., and N.K.; Analysis and/or interpretation of data: A.B., H.F.S., M.S., and C.M.M.; Drafting of the manuscript: A.B. and H.F.S.; All authors revised the manuscript critically for important intellectual content.

## Competing interests

The authors declare no competing interests.
