## [Transparent Peer Review File. · Communications Biology]

Reviewers' comments:

Reviewer #1 (Remarks to the Author):

This study examined the spatial-temporal dynamics of large-scale brain networks in young children with autism spectrum disorders and typically developing control (TDC) children. Compared with the TDC group, children with ASD showed significantly decreased frequency of occurrence in microstate class C. Moreover, the occurrence of microstate class C is negatively associated with symptom severity of ASD. Overall, the paper is well-organized; the methods and results are presented in a clear way. However, several concerns should be addressed before it can be considered for publication.

1. Both male and female children were included in the analysis. However, an increasing number of studies have reported the sexual differentiation on brain structure and function in ASD (e.g., doi: 10.1093/brain/awt216; doi: 10.1093/scan/nsw027). Therefore, it may be interesting to explore whether there is any difference in the spatial-temporal dynamics of brain networks between males and females with ASD.
2. Intellectual ability is also regarded as one important factor that contributes to the heterogeneity of ASD (e.g., doi: 10.1038/s41380-019-0420-6). Therefore, it is strongly recommended to consider the effects of intelligence in studies of autism. Has this study measured intelligence quotient scores for participants? How did the authors exclude potential intelligence effects on brain networks?
3. Given that this study aimed to explore the temporal dynamics of brain networks, I suggest to report and compare the temporal parameters for each microstate instead of a sum of these parameters. In addition, did all participants have five microstates in the microstate analysis? If not, how many participants were there in each microstate?
4. The authors used a k-means cluster analysis, which is a data-driven approach, to obtain the microstates. From a methodological perspective, that is fine. However, it is difficult to associate the cognitive/clinical profiles of ASD with these microstates. For example, one good way to explain the potential psychological and physiological processes of each microstate is meta-analytic decoding (doi: 10.1002/hbm.24447). So I suggest acknowledging this as a limitation of this study.
5. What does the color bar in Figure 1 represent? What is the range of value of it?
6. Please add a column of statistics (e.g. p values) comparing demographic variables between ASD and TDC groups in Table 1.

Reviewer #2 (Remarks to the Author):

2020 Bochet Review Comments

Summary

The authors examine differences in the spatiotemporal dynamics of whole-brain neuronal networks in a large sample of preschoolers with Autism Spectrum Disorders (ASD) and age-matched typically developing (TD) peers.

The main findings are as follows:

1. The authors identified five template maps (A-E) which explained 78.5% of the variance in the data.
2. The topographies of maps A-D are said to be identical to the canonical microstate maps. Map E is said to be topographically similar to map E from two other studies.
3. The summed time coverage and summed occurrence are different between the two groups.
4. The difference in summed occurrence is driven by a decrease in the occurrence of microstate C in the ASD group.
5. The authors found that the decrease in microstate C occurrence showed a negative and a positive correlation with ADOS severity and a subscale of the VABS-II, respectively.

The authors conclude that:

1. The default mode network (DMN) represented by microstate C was activated less frequently in children with ASD.
2. Greater decrease in the activity of the DMN measured as a decrease in microstate C occurrence mediated more severe ASD symptoms and worse adaptive skills.

General impressions:

The findings reported in this study as novel in terms of the age range of the population studied. Similar findings have been reported between ASD and TD children in older age groups – Jia et al., 2018 and Nagabhushan Kalburgi et al., 2020. The findings of the paper under review are in agreement with the overall conclusions of these two other previously published studies. Another article by D’Croz Baron et al., 2019 in adults with and without ASD found results that do not agree with the manuscript under review.

The author’s rigor and transparency on the inclusion criteria for proper netting, excluding noisy data due to artefact, etc. assures me that the findings are indeed real and not an artifact of the difference in quality of data between the ASD and TD groups. It is very difficult to collect good quality data from toddlers. The author’s transparency about exclusion of >50% of the participants due to data quality is a plus in this case.

The findings from this study show that the brain networks are altered at a very young age in individuals with ASD. There is potential in using these brain network changes as a marker of treatment/intervention efficacy.

Major concerns:

1. The authors mention that the differences in the theta and alpha frequency band activities in various brain regions gave impetus to conducting microstate analysis on EEG data from ASD and TD toddlers.
 - a. What is the relationship between the theta and alpha frequency band differences and microstate differences observed?
 - b. Do they represent the same activities?
 - c. Are microstates frequency specific?
 - d. It would be great if the authors could tie the information presented in the introduction together with the results from this study in the discussion section.
2. The occurrence of the summed microstates as well as the individual microstate occurrences are off by a factor of 100 from other published literature. This is very concerning. Do the authors have an explanation for this?
3. What is the statistical model? The norm in the field is to use a 2x5 ANOVA with 2 groups (ASD and TD) and 5 microstates (A, B, C, D, E) as independent variables and the parameters being the dependent measures. Could the authors provide their rationale for conducting unpaired t-tests of the summed parameters and then unpaired t-tests at each microstate level?
4. What was the minimum number of clean epochs required per participant to be considered sufficient?
 - a. Were the same number of epochs chosen across all participants?
 - b. If so, were these randomly chosen or were the first x many epochs included in analyses?
5. Could the authors provide a table with the mean and standard deviations of the duration, occurrence, coverage and GEV values by group and microstate?
6. The discussion states “The meta-criterion determined an optimal number of five template maps that best described the entire dataset explaining 78.5 percent of the global variance.” However, in Figure 2, the average GEV of the two groups appears to be less than 60%. Do the authors have an explanation for this seemingly discordant information?

7. The authors state that, "The time coverage is the percentage of total time for a given cluster map in the individual EEG recording."

- a. So, for each participant, total time can be divided up into time spent in A, B, C, D, E or the remaining unassigned topography states. The total time should add up to 100%.
- b. If the authors found a difference for the summed time coverage, but not when the microstates are looked at separately, then is the coverage of the unassigned topographies driving these differences? Wouldn't this argue that the GEVs are not comparable?
- c. Please explain.

8. The Spearman correlation is weak for microstate C and ADOS total severity score ($r = -0.359$, $N = 47$, $p = 0.013$) and daily living skills subdomain of the Vineland Adaptive Behavior Scale-II (VABS-II) ($r = 0.396$, $N = 47$, $p = 0.006$).

- a. What other brain-behavior correlations were measured?
- b. How many comparisons were made?
- c. Were these corrected for multiple comparisons?
- d. In the results section, the authors state that there is a positive association between microstate C occurrence and ADOS severity with an $r^2 = -0.359$. In the same paragraph, the authors state that there is also a positive association between the occurrence of microstate C and the daily living skills subdomain of the Vineland Adaptive Behavior Scale-II with an $r^2 = 0.396$. I believe one of these should be a negative association. Please check these statements such that the wording is consistent with the findings.

9. The authors state that the microstates template maps obtained in this study were identical to the canonical maps.

- a. How was the similarity of the microstates A-E in this study compared to those reported in other studies?
- b. Was dissimilarity calculated or was it determined by visual comparison?
- c. Jia et al., 2019 and Nagabhushan Kalburgi et al., 2020 have reported unique microstate topographies for microstate D. Were there any group level differences in the template microstate maps in this study?

10. Were verbally fluent children with ASD recruited? Please provide subscale scores of the Mullen Scales of Learning and the Vineland Adaptive Behavior Scale-II.

Minor Comments:

1. The final sample is not gender matched. Please comment.
2. Out of the 150 toddlers with successful EEG netting, were there differences in how reliably the net was placed across the participants, i.e. were any participants excluded because the cap was not accurately placed?
3. Parameters studied include the GEV, the mean duration, the frequency of occurrence and the time coverage. If you are studying directed brain network connectivity, why was transition probability/syntax not analyzed?

Reviewer #1 (Remarks to the Author):

Summary : This study examined the spatial-temporal dynamics of large-scale brain networks in young children with autism spectrum disorders and typically developing control (TDC) children. Compared with the TDC group, children with ASD showed significantly decreased frequency of occurrence in microstate class C. Moreover, the occurrence of microstate class C is negatively associated with symptom severity of ASD. Overall, the paper is well-organized; the methods and results are presented in a clear way. However, several concerns should be addressed before it can be considered for publication.

Q1. Both male and female children were included in the analysis. However, an increasing number of studies have reported the sexual differentiation on brain structure and function in ASD (e.g., doi: 10.1093/brain/awt216; doi: 10.1093/scan/nsw027). Therefore, it may be interesting to explore whether there is any difference in the spatial-temporal dynamics of brain networks between males and females with ASD.

We thank the reviewer for this relevant comment and pertinent references. We did not conduct separate analyses for each gender as we believe it is valuable to include both genders in the study as well as preserve our sample size. Unfortunately, the number of females with ASD in our study is small ($n = 11$). Taking into account the literature on sex differences, we have now better balanced our groups so that they contain an equivalent proportion (20%) of females, in order to minimize any gender bias. We then conducted a male-only analysis (removed 19 female subjects (11 ASD, 8 TD)) as a validation of our attempt to control for gender bias. The findings from the male-only analysis indicated that group differences for map B were preserved and very similar to the larger analysis, indicating that the larger analysis is not biased by gender.

In the revised manuscript, the following modifications were made:

Section METHOD, subsection Participants "In total, 293 participants were recruited for the experiment. We did not manage to place the cap on 108 participants. The cap was placed on the heads of 185 participants (110 ASD and 75 TD). Out of those, 60 participants (44 ASD and 16 TD) were excluded due to having too many movement-related artifacts, noisy signal, lack of interest or having insufficient number of epochs for subsequent analysis. This was to be expected given the sensory processing issues frequently reported in children with ASD (Kojovic, 2019). In order to minimize gender bias and include an equivalent percentage (20%) of females in both groups, we excluded 12 females from the TD group. As a result, the final sample consisted of 113 participants: 66 toddlers and preschoolers with ASD (11 females; mean age 3.3 years \pm 1.0, range 1.8-5.9) and 47 TD peers (8 females; mean age 3.3 years \pm 1.2, range 1.8-5.8). Groups did not differ by age ($p = 0.8838$) or sex ($p > 0.9999$)."

Section METHOD, subsection Statistical Analysis "We first checked if the temporal parameters of the microstate classes had a normal distribution using D'Agostino and Pearson's tests. We investigated group differences using unpaired t-tests or Mann-Whitney-Wilcoxon pairwise tests if data were not normally distributed for each temporal parameter of each map. We applied False Discovery Rate (FDR) correction for multiple comparisons taking into account the four parameters of the five maps. We then conducted a male-only analysis (removed 19 female subjects (11 ASD, 8 TD)) as a validation of our attempt to control for gender bias. The findings from the male-only analysis were very similar to the larger analysis, indicating that the larger analysis is not biased by gender."

Section RESULTS, subsection Microstate analysis "When looking for group differences in temporal parameters for the five maps, we found major differences in the temporal parameters of map B between toddlers and preschoolers with ASD and their TD peers (Figure 3). All four parameters - GEV, mean duration, time coverage and occurrence - of the map B were significantly increased in toddlers and preschoolers with ASD in comparison to their TD peers (GEV : $p < 0.001$; Mean Duration : $p < 0.001$; Time Coverage : $p < 0.001$; Occurrence : $p < 0.001$). These differences survived FDR correction. As already mentioned, these differences in map B parameters also persisted when all females (11 ASD, 8 TD) were removed from the analysis (GEV : $p < 0.001$; Mean Duration : $p < 0.001$; Time Coverage : $p < 0.001$; Occurrence : $p < 0.001$)."

Q2. Intellectual ability is also regarded as one important factor that contributes to the heterogeneity of ASD (e.g., doi: 10.1038/s41380-019-0420-6). Therefore, it is strongly recommended to consider the effects of intelligence in studies of autism. Has this study measured intelligence quotient scores for participants? How did the authors exclude potential intelligence effects on brain networks?

We thank the reviewer for their comment. Indeed, differences in cognitive skills are part of the heterogeneity found in autism. Considering the young age of participants, cognitive functioning was assessed using two standardized tools: the Mullen Scales of Early Learning composite score (MSEL) (Mullen, 1995), or the Psycho-Educational Profile, third edition (PEP-3) (Schopler et al., 2005). The MSEL is a standardized assessment used to measure cognitive functioning for children from birth through age 68 months. Information about cognitive functioning includes five subdomains: visual reception, fine and gross motor skills, receptive and expressive language. The total development quotient (total DQ) is referred to as an estimate of overall intelligence. In comparison, the PEP-3 is a standardized tool to assess developmental level of children with developmental disorders, in particular ASD, between 2 and 7.5 years old. It includes the same five domains of the MSEL but combines nonverbal and verbal intelligence scores into the same verbal and preverbal cognition scale and adds an Imitation skills index. We used either the MSEL or the PEP-3, depending when the children were included in our protocol. We indeed added the MSEL to our standardized battery for cognitive assessment only after 2015. As a result, in the current study, cognitive skills for 55 children with ASD were estimated using the MSEL, and for 10 children with ASD using the Verbal/Preverbal Cognition scale of the PEP-3. Data on cognitive skills were missing for one child.

In total, 37 of 65 children with ASD (56 percent, $F = 5$) obtained a score equal to or above 70 (range 71.0 – 132.2). Conversely, 28 of 65 children with ASD (44 percent, $F = 6$) obtained a score below 70, (range 21.9 – 69.7).

To explore the effect of cognitive skills, we correlated between cognitive skills data and temporal parameters of the maps. This correlation analysis included 55 children with ASD with MSEL scores available (total DQ and subdomains). Indeed, we did not wish to mix data from different tests (MSEL and PEP-3) to avoid inducing bias.

There was no correlation between total DQ of the MSEL with any of the temporal parameters of maps A to E in children with ASD (p -values ranging from 0.093 to 0.973).

In the revised manuscript, the following modifications were made:

Section METHOD, subsection Participants "Considering the young age of participants, cognitive functioning was assessed using one of two standardized tools: the Mullen Scales of Early Learning composite score (MSEL) (Lee, 2013) or the Psycho-Educational Profile, third edition (PEP-3) (Schopler et al., 2005). The MSEL is a standardized assessment used to measure cognitive functioning for children from

birth through age 68 months. Information about cognitive functioning includes five subdomains: visual reception, fine and gross motor skills, receptive and expressive language. The total development quotient (total DQ) is referred to as an estimate of overall intelligence. In comparison, the PEP-3 is a standardized tool to assess developmental level of children with developmental disorder, in particular ASD, between 2 and 7.5 years old. It includes the same five domains of the MSEL but combines nonverbal and verbal intelligence scores into the same verbal and preverbal cognition scale and adds an imitation skills index. We used either the MSEL or the PEP-3, depending when the children were included in our protocol. The MSEL was added to our standardized battery for cognitive assessment only after 2015. As a result, in the current study, cognitive skills for 55 children with ASD were estimated using the MSEL, and for 10 children with ASD, using the verbal/preverbal cognition scale of the PEP-3 in. Data on cognitive skills were missing for one child with ASD.”

Section RESULTS, subsection Correlation with clinical measures "The results of Pearson's correlations and Spearman's rank correlations suggested specific microstate classes were related to specific symptoms in the toddlers and preschoolers with ASD group. Indeed, when looking at ASD symptoms severity, we found a negative correlation between ADOS social affects severity score and the mean duration of the map E (uncorrected p-value = 0.030). This suggests that children with more social difficulties, have a shorter duration of the map E. Considering the developmental level of children with ASD, we found negative correlations between the fine motor domain of the MSEL and three temporal parameters of the map D - GEV (non-corrected p-value = 0.006), time coverage (non-corrected p-value = 0.007) and occurrence (non-corrected p-value = 0.005). This suggests that the lower the child's fine motor skills are, the less prevalent their map D. Finally, considering the comorbid symptoms of children with ASD, we mostly found positive correlations between children's affective problems reported in CBCL 1.5-5 and three temporal parameters of the map B - GEV (non-corrected p-value = 0.003), time coverage (non-corrected p-value = 0.008) and occurrence (non-corrected p-value = 0.048). This suggests that the more emotional difficulties the child has, the more prevalent map B is. We also found negative correlation between children's attention deficits and hyperactivity problems reported in CBCL 1.5-5 and the mean duration of the map C (non-corrected p-value = 0.020), suggesting that the more attention deficits and hyperactivity problems the child has, the shorter the duration of the map C is. However, correlations were not significant after FDR correction for multiple comparisons, and hence should be considered exploratory. Results of the exploratory correlations are reported in Figure 4.”

Q3. Given that this study aimed to explore the temporal dynamics of brain networks, I suggest to report and compare the temporal parameters for each microstate instead of a sum of these parameters. In addition, did all participants have five microstates in the microstate analysis? If not, how many participants were there in each microstate?

We thank the reviewer for their comment and questions. As mentioned above, this comment helped us to highlight two major errors in the script of the EEG software used to estimate temporal parameters. The errors we found were: 1) the units of the temporal parameters extracted from the microstate maps were not incorrect, and 2) artifacts were not excluded from the analysis in the last step of the procedure (back-fitting). Both errors have induced biases and inconsistencies. We have now rectified these errors and re-analyzed the data with the newly released version of the EEG software. We agreed that it would not be useful nor appropriate to compare the sum of temporal parameters as for certain parameter such as time coverage, the sum would equal to 100 percent for each participant. In the revised manuscript we therefore directly compared the temporal parameters for each microstate instead of a sum of these parameters.

To address the question about the number of microstates for each participant, we used an established clustering method in the EEG literature, as detailed in the Figure 7 of the revised manuscript. First, for each individual resting-state recording, peaks of GFP were determined and their specific potential maps were selected. Then, we applied a first k-means clustering procedure using these potential maps. For each subject, we obtained a best k-means clustering solution based on the meta-criterion. These best solutions varied between 4 to 9 maps for each subject in the TD group and between 3 to 10 maps for each subject in the ASD group. The best solutions obtained at individual level were submitted to a second k-means group cluster analysis. We obtained a best solution of 5 microstate classes, using the meta-criterion. Then, we used the best solutions obtained at individual level separately for ASD and TD subjects and we extracted the 5 microstate classes template each group. The 5 maps templates were highly similar across both groups, according to spatial correlation applied to each pair of microstate classes (Pearson's spatial correlation coefficients for maps A ASD vs TD = 0.992, maps B ASD vs TD = 0.986, maps C ASD vs TD = 0.996, maps D ASD vs TD = 0.958, maps E ASD vs TD = 0.985). Following this, we used the 5 template maps extracted from the segmentation of the ASD group to fit it back in the original EEG recordings of the 66 children with ASD, and the 5 template maps extracted from the segmentation of TD group to fit it back in original EEG recordings of the 47 TD children. Each time point is labeled with the cluster map of the template having the highest spatial correlation. This way, we obtained the temporal parameter values for the 5 maps for all 113 participants.

In the revised manuscript, the following modifications were made:

Section METHOD, subsection Microstate analysis "The pipeline for the analysis is illustrated in (Figure 7). We applied a k-means cluster analysis to the data of each subject to estimate the optimal set of topographies explaining the EEG signal. The clustering was applied only at local maxima of the Global Field Power (GFP) which is calculated as the standard deviation of all electrodes at a given time point and represents time points of highest signal-to-noise ratio (see (Murray et al., 2008) for formulas). The polarity of the maps was ignored during the procedure. The k-means cluster analysis was first computed at the individual level and then across all participants (children with ASD and TD children together) to obtain the group cluster maps. In order to determine the optimal number of maps at both levels (both within and across subjects), we applied a meta-criterion that includes seven independent criteria. For a detailed description of these criteria, see (Custo et al., 2017) and (Bréchet et al., 2019). Then we identified the five prototypical maps that best described the dataset of toddlers and preschoolers with ASD and TD peers separately. We computed Pearson's spatial correlation for each of the five microstate classes to ensure that those five microstate classes were similar across both groups. High spatial correlation coefficients indicated that the microstate classes were similar amongst both groups and the five cluster maps extracted from both groups separately were then fitted back to the original EEG of each subject, depending on their group. This way, spatial correlation was calculated between the cluster maps and each individual data point and that data point was labeled with the cluster map that showed the highest correlation. Polarity of the maps was again ignored for the back-fitting procedure. Data points that did not correlate more than 50 percent with a given group cluster map were marked as unlabeled. Only labeled data points were included in the analysis, meaning that unlabeled data points and periods marked as artifacts during the preprocessing were excluded."

Section RESULTS, subsection Microstate analysis "The k-means cluster analysis across all participants identified five dominant maps, which explained 80.1 percent of the total variance (Figure 1).

The topographies of these five cluster maps corresponded to the canonical microstate classes previously reported in the literature and were labeled accordingly (map A, B, C, D and E) (Michel and König, 2018). We also applied a k-means cluster analysis across toddlers and preschoolers with ASD and TD peers separately and extracted the five cluster maps for both groups. For both groups, the five cluster maps also corresponded to the canonical microstate classes previously reported in the literature and were therefore labeled accordingly (map A, B, C, D and E). The five cluster maps extracted from ASD group explained 80.8 percent of the total variance of EEG recordings of this group and the five cluster maps extracted from TD group explained 81.9 percent of the total variance of EEG recordings of this second group. The five cluster maps extracted from both groups separately were highly similar (Pearson's spatial correlations coefficients for map A = 0.992, map B = 0.986, map C = 0.996, map D = 0.958, map E = 0.985) and were used for further analysis (Figure 2)."

Q4. The authors used a k-means cluster analysis, which is a data-driven approach, to obtain the microstates. From a methodological perspective, that is fine. However, it is difficult to associate the cognitive/clinical profiles of ASD with these microstates. For example, one good way to explain the potential psychological and physiological processes of each microstate is meta-analytic decoding (doi: 10.1002/hbm.24447). So I suggest acknowledging this as a limitation of this study.

We thank the reviewer for this comment and the interesting reference. Indeed, fMRI is a very good way to study brain network dynamics and correlation with the clinical data because of the technique's high spatial resolution. In comparison to fMRI, EEG has poor spatial resolution, but the excellent temporal resolution provided by EEG allows us to show the alteration of the "micro-dynamics" of these networks at a sub-second scale. Moreover, EEG is a tool that is more easily used with toddlers and preschoolers with ASD. In our revised manuscript, we acknowledge the low spatial resolution of EEG as a limitation in the discussion and we cite the reference proposed by the reviewer.

In the revised manuscript, we modified the DISCUSSION as follows:

"Resting-state brain networks (RSNs) are usually studied using functional MRI (fMRI) because of its high spatial resolution (Lee et al., 2013). Guo and colleagues used fMRI technique to study the brain dynamic connectivity and brain-behaviors relationships in ASD and they found partially impaired functional connectivity states between right anterior insula and default mode network (Guo et al., 2019). However, these RSNs are sensitive to dynamic fluctuation (Prete et al., 2017), which are difficult to capture with fMRI because of its moderate temporal resolution (in the order of seconds) and the delayed hemodynamic response. As such, EEG is a valuable alternative technique to study RSNs owing to its sub-second scale temporal resolution (Abreu et al., 2020; Michel and König, 2018). The increased prevalence of microstate class B in toddlers and preschoolers with ASD could reflect the atypical inter-network connectivity between VIS and other majors RSNs amongst individuals with ASD reported in the literature. For example, Gao and colleagues suggested hyper-connectivity between VIS and default mode network (DMN) probably impacting on language abilities in ASD (Gao et al., 2019). Morgan and colleagues also highlighted hyper-connectivity between VIS, DMN and language network (Morgan et al., 2019)."

Q5. What does the color bar in Figure 1 represent? What is the range of value of it?

We thank the reviewer for pointing out this missing information. The color bar in Figure 1 represents the electric field power, in μV . However, following the reviewer's comment, we realize that this color bar is more of a distraction than a source of important information. It is

rarely reported in other microstate studies and so we decided to remove it from Figure 1 in the revised manuscript.

Q6. Please add a column of statistics (e.g. p values) comparing demographic variables between ASD and TDC groups in Table 1.

We thank the reviewer for highlighting this missing information. In the revised manuscript, in Table 1., we added a column of statistics comparing demographic variables between groups.

Reviewer #2 (Remarks to the Author):

Summary : The authors examine differences in the spatiotemporal dynamics of whole-brain neuronal networks in a large sample of preschoolers with Autism Spectrum Disorders (ASD) and age-matched typically developing (TD) peers.

The main findings are as follows:

1. The authors identified five template maps (A-E) which explained 78.5% of the variance in the data.
2. The topographies of maps A-D are said to be identical to the canonical microstate maps. Map E is said to be topographically similar to map E from two other studies.
3. The summed time coverage and summed occurrence are different between the two groups.
4. The difference in summed occurrence is driven by a decrease in the occurrence of microstate C in the ASD group.
5. The authors found that the decrease in microstate C occurrence showed a negative and a positive correlation with ADOS severity and a subscale of the VABS-II, respectively.

The authors conclude that:

1. The default mode network (DMN) represented by microstate C was activated less frequently in children with ASD.
2. Greater decrease in the activity of the DMN measured as a decrease in microstate C occurrence mediated more severe ASD symptoms and worse adaptive skills.

General impressions: The findings reported in this study as novel in terms of the age range of the population studied. Similar findings have been reported between ASD and TD children in older age groups – Jia et al., 2018 and Nagabhushan Kalburgi et al., 2020. The findings of the paper under review are in agreement with the overall conclusions of these two other previously published studies. Another article by D'Croz Baron et al., 2019 in adults with and without ASD found results that do not agree with the manuscript under review.

The author's rigor and transparency on the inclusion criteria for proper netting, excluding noisy data due to artefact, etc. assures me that the findings are indeed real and not an artifact of the difference in quality of data between the ASD and TD groups. It is very difficult to collect good quality data from toddlers. The author's transparency about exclusion of >50% of the participants due to data quality is a plus in this case.

The findings from this study show that the brain networks are altered at a very young age in individuals with ASD. There is potential in using these brain network changes as a marker of treatment/intervention efficacy.

Major concerns:

Q1. The authors mention that the differences in the theta and alpha frequency band activities in various brain regions gave impetus to conducting microstate analysis on EEG data from ASD and TD toddlers.

- a. What is the relationship between the theta and alpha frequency band differences and microstate differences observed?
- b. Do they represent the same activities?
- c. Are microstates frequency specific?
- d. It would be great if the authors could tie the information presented in the introduction together with the results from this study in the discussion section.

We thank the reviewer for his comment and questions. Upon rereading the introduction, we acknowledge that we were not clear enough regarding the purpose of the present study. We thought it would be interesting to look at our EEG data using another approach, that is - microstate analysis and with a bigger sample (the initial study with the functional connectivity method only included 36 participants compared to the current one that includes 113 participants). As the reviewer mentions, acquiring data in toddlers is a rather difficult but important endeavor. The literature indicates that microstates' topographies are spatially equivalent across all frequencies, matching the canonical broadband maps (A, B, C, and D) (See Britz et al. 2010). It is a surface broadband approach where all frequencies between 1-40 Hz are included and as such not directly comparable to our previously published study where we used a directed functional connectivity in the source space and where differences appear in specific frequency bands. We have now adapted the introduction to clarify this aspect.

In the revised manuscript, we modified the last paragraph of the INTRODUCTION as follows:

"Here, we hypothesized that the toddlers and preschoolers with ASD would show differences in the spatio-temporal properties as well as transition probabilities of some microstates compared to their TD peers. Considering the large heterogeneity in ASD in terms of autism symptom severity, developmental level and possible comorbid symptoms, we investigated brain-behavioral relationships between the temporal characteristics of the microstates and clinical phenotype. Finally, we used a bootstrapping approach (Schaer et al., 2015) to examine the stability of our findings. Post-hoc power analyses depending on the observed effect sizes were made to estimate the relationship between the sample size of our group and the observed statistical power. The bootstrapping procedure served to estimate the likelihood of finding the true result we observed in our full cohort from smaller sample sizes of participants."

Q2. The occurrence of the summed microstates as well as the individual microstate occurrences are off by a factor of 100 from other published literature. This is very concerning. Do the authors have an explanation for this?

We thank the reviewer for this comment and for raising this issue. Indeed, as mentioned above, this comment helped us to highlight two major errors in the script of the EEG software used to estimate temporal parameters. The errors we found were: 1) the units of the temporal parameters extracted from the microstate maps were not incorrect, and 2) artifacts were not excluded from the analysis in the last step of the procedure (back-fitting). Both errors have induced biases and inconsistencies. We have now rectified these errors, re-analyzed the data with the newly released version of the EEG software, and obtained values of occurrences that are now coherent with the published literature (around 1-2 occurrence/s for a map).

Q3. What is the statistical model? The norm in the field is to use a 2x5 ANOVA with 2 groups (ASD and TD) and 5 microstates (A, B, C, D, E) as independent variables and the parameters being the dependent measures. Could the authors provide their rationale for conducting unpaired t-tests of the summed parameters and then unpaired t-tests at each microstate level?

We thank the reviewer for this comment and question. As mentioned in our answer to Q2, we have identified errors in the EEG software script, rectified these and reanalyzed the data. We agreed that it would not be useful nor appropriate to compare the sum of temporal parameters as for certain parameter such as time coverage, the sum would equal to 100 percent for each participant. In the revised manuscript we therefore directly compared the temporal parameters for each microstate instead of a sum of these parameters.

We applied unpaired t-tests or Mann-Whitney tests between the ASD and TD groups, depending on whether temporal parameter values were normally distributed or not, respectively. We corrected for multiple comparisons with the False Discovery Rate (FDR) method. We do not to apply ANOVA, as we only have one independent variable which is the diagnosis (ASD versus TD). In the literature, Damborska and colleagues (Damborska 2019a, 2019b) also applied repeated unpaired t-tests and corrected with FDR to compare temporal parameters of maps between depressive or bipolar patients and controls. Jia and Yu applied ANCOVA to compare temporal parameters of maps between children with ASD and TD children, and to covariate temporal parameters for age. However, they have a larger age range than us (5-18 y.o. instead of 1-5 y.o.). To ensure that we have no effect of age on the results, we checked by correlating the age of participants with temporal parameters and there was no significant correlation (p-values range from 0.096 to 0.966).

In the revised manuscript, the following modifications were made:

Section METHOD, subsection Statistical analysis "We first checked if the temporal parameters of the microstate classes had a normal distribution using D'Agostino and Pearson's tests. We investigated group differences using unpaired t-tests or Mann-Whitney-Wilcoxon pairwise tests if data were not normally distributed for each temporal parameter of each map. We applied False Discovery Rate (FDR) correction for multiple comparisons taking into account the four parameters of the five maps. We then conducted a male-only analysis (removed 19 female subjects (11 ASD, 8 TD)) as a validation of our attempt to control for gender bias. The findings from the male-only analysis were very similar to the larger analysis, indicating that the larger analysis is not biased by gender."

To ensure that age had no effect on the results, we correlated the age of participants with temporal parameters, applying two-tailed Pearson's correlation or two-tailed Spearman's rank correlation, depending on the normality of distribution of the data."

Section RESULTS, subsection Microstate analysis "There was no effect of age on the results, as correlations between temporal parameters and age's participants were not significant (p-values range from 0.096 to 0.966)."

Q4. What was the minimum number of clean epochs required per participant to be considered sufficient?

a. Were the same number of epochs chosen across all participants?

b. If so, were these randomly chosen or were the first x many epochs included in analyses?

We thank the reviewer for these questions. We recorded approximately 5 minutes of spontaneous EEG for each participant. In our case, 1 time frame (or epoch) = 8ms of recording. During the preprocessing procedure, we only excluded bad epochs (i.e. artifacts) and EEG recordings with too many artifacts were excluded from the study. We did not apply a strict requirement for a particular number of clean epochs per participants, as there is no specific guideline regarding this in the literature. In total, the mean of recorded epochs for ASD group was 40308 (range 27149 - 53161) and for TD groups was 39326 (21092 - 48825). After the fitting procedure for both groups, all recordings had a percentage of

"labeled" epochs higher than 50 percent meaning that at minimum half of the recorded epochs were labeled as one of the 5 microstate maps. The bad epochs marked as artifacts during the preprocessing procedure were considered as "unlabeled" epochs. In total, the mean of labeled epochs for ASD group was 33239 (range 17752 - 44394) and for TD groups was 34110 (15514 - 43074). Temporal parameters of microstate maps were then computed only on the labeled epochs, meaning that we have on average more than 4 minutes of recording per participant that were analyzed.

Kanna and colleagues showed high test-retest reliability and cross-method consistency of microstate features using 3 minutes of resting-state EEG recording (Kanna et al., 2014). Moreover, Liu and colleagues showed no significant main effect for microstates temporal parameters and transition probabilities applying repeated measures one-way ANOVA of the microstates' characteristics among various epoch durations (from 0.5 to 5 min) (Liu et al., 2020).

Q5. Could the authors provide a table with the mean and standard deviations of the duration, occurrence, coverage and GEV values by group and microstate?

We thank the reviewer for pointing out this missing information.

In the revised manuscript, we added Table 2 with the four temporal parameters values for each map.

Q6. The discussion states "The meta-criterion determined an optimal number of five template maps that best described the entire dataset explaining 78.5 percent of the global variance." However, in Figure 2, the average GEV of the two groups appears to be less than 60%. Do the authors have an explanation for this seemingly discordant information?

We thank the reviewer for this comment. As mentioned in the answers to Q2 and Q3, we have reanalyzed the data following the discovery of two major errors in the script of the EEG software we used to estimate temporal parameters.

In the revised manuscript, we corrected the units of temporal parameters. Indeed, the Global Explained Variance as a temporal parameter is not a percentage but estimated value between [0 : 1] of the explained variance of a given map, weighted by the GFP (for more details about the involved mathematical formulae, see (Murray et al., 2008)). This is different from the global variance of the five microstate classes' templates, which is a percentage of the explained variance of the five microstate maps for the dataset.

Q7. The authors state that, "The time coverage is the percentage of total time for a given cluster map in the individual EEG recording."

- a. So, for each participant, total time can be divided up into time spent in A, B, C, D, E or the remaining unassigned topography states. The total time should add up to 100%.**
- b. If the authors found a difference for the summed time coverage, but not when the microstates are looked at separately, then is the coverage of the unassigned topographies driving these differences? Wouldn't this argue that the GEVs are not comparable?**
- c. Please explain.**

We thank the reviewer for this comment and question. As mentioned in the answers to Q2, Q3 and Q6, we have reanalyzed the data following the discovery of two major errors in the script of the EEG software we used to estimate temporal parameters.

In the revised manuscript, the dynamic parameters are computed only on the labeled timeframes and the sum time coverage for 5 maps for each participant is indeed equal to 100 percent. There is therefore no longer an interest to compare sum time coverage between groups.

Q8. The Spearman correlation is weak for microstate C and ADOS total severity score ($r = -0.359$, $N = 47$, $p = 0.013$) and daily living skills subdomain of the Vineland Adaptive Behavior Scale-II (VABS-II) ($r = 0.396$, $N = 47$, $p = 0.006$).

a. What other brain-behavior correlations were measured?

b. How many comparisons were made?

c. Were these corrected for multiple comparisons?

d. In the results section, the authors state that there is a positive association between microstate C occurrence and ADOS severity with an $r^2 = -0.359$. In the same paragraph, the authors state that there is also a positive association between the occurrence of microstate C and the daily living skills subdomain of the Vineland Adaptive Behavior Scale-II with an $r^2 = 0.396$. I believe one of these should be a negative association. Please check these statements such that the wording is consistent with the findings.

We thank the reviewer for these questions and comments. Indeed, correlations were weak and as we had to redo all analyses in the revised manuscript, we decided to interpret correlations between brain-behavioral relationships and temporal parameters of the microstate classes in an exploratory manner.

In the revised manuscript, we focused correlation analyses on three behavioral measures to explore more specifically the heterogeneity of toddlers and preschoolers with ASD: autism symptom severity, developmental level and comorbid symptoms. We explored relationships between temporal parameters and autism symptoms severity with the Autism Diagnostic Observation Schedule total, social affect and repetitive and restricted interest severity scores (ADOS-G and ADOS-2) (Lord et al., 2000; Lord et al., 2012). We then explored the relationships with developmental level of children, using the Mullen Scales of Early Learning composite score (MSEL) (Lee, 2013) with the total developmental quotient and fine motor skills, receptive and expressive language, and visual reception subdomains scales. We also explored the relationships with comorbid symptoms using the five DSM-scales of the Child Behaviors Checklist 1.5-5 years old version parents' questionnaire (Achenbach & Rescorla, 2000). We applied False Discovery Rate (FDR) correction to correct for multiple correlations taking into account the four parameters of the five maps and the three different clinical evaluations we used (ADOS, MSEL, CBCL). In total, we performed 60 comparisons with the ADOS (3 ADOS severity scores x 4 parameters x 5 maps), 100 comparisons with the MSEL (5 MSEL DQ x 4 parameters x 5 maps) and 100 comparisons with the CBCL 1.5-5 (5 CBCL T-scores x 4 parameters x 5 maps). We applied False Discovery Rate (FDR) correction to correct for multiple correlations taking into account the four parameters of the five maps and the different clinical evaluations we used (ADOS, MSEL, CBCL); however, no correlation survived correction and hence this analysis was considered exploratory.

In the revised manuscript, the following modifications were made:

Section METHOD, subsection Statistical analysis "Considering the large heterogeneity in ASD, we investigated possible brain-behavioral relationships between the temporal parameters of the microstate classes with children phenotype. Among toddlers and preschoolers with ASD, we correlated autism symptom severity, developmental level and comorbid symptoms with all temporal parameters.

We applied two-tailed Pearson's correlation or two-tailed Spearman's rank correlation depending on the normality of distribution of the data between the GEV, mean duration, time coverage and occurrence of microstates A, B, C, D and E and Autism Diagnostic Observation Schedule total, social affect and repetitive and restricted interest severity scores (ADOS-G and ADOS-2) (Lord et al., 2000; Lord et al., 2012). To explore the relationship between temporal parameters and the developmental level of children, we also applied correlation with the Mullen Scales of Early Learning composite score (MSEL) (Lee, 2013), using the total developmental quotient and fine

motor skills, receptive and expressive language, and visual reception subdomains scales. We also explored comorbid symptoms by correlating temporal parameters with the five DSM-scales of the Child Behaviors Checklist 1.5-5 years old version parents' questionnaire (Achenbach & Rescorla, 2000). The five DSM-scales are Attention Deficit/Hyperactivity Problems, Anxiety Problems, Affective Problems, Oppositional Defiant Problems and Pervasive Developmental Problems. We applied False Discovery Rate (FDR) correction to correct for multiple correlations taking into account the four parameters of the five maps and the different clinical evaluations we used (ADOS, MSEL, CBCL)."

Section RESULTS, subsection Correlations with clinical measures "The results of Pearson's correlations and Spearman's rank correlations suggested that specific microstate classes are related with specific symptoms in the toddlers and preschoolers with ASD group. Indeed, when looking at ASD symptoms severity, we found mostly a negative correlation between ADOS social affects severity score and the mean duration of the map E (non-corrected p-value = 0.030). This suggests that the more social difficulties the child has, the shorter the duration of the map E is. Considering the developmental level of children with ASD, we found negative correlations between the fine motor domain of the MSEL and three temporal parameters of the map D - GEV (non-corrected p-value = 0.006), time coverage (non-corrected p-value = 0.007) and occurrence (non-corrected p-value = 0.005). This suggests that the lower the child's fine motor skills are, the less prevalent the map D is. Finally, considering the comorbid symptoms of children with ASD, we mostly found positive correlations between children's affective problems reported in CBCL 1.5-5 and three temporal parameters of the map B - GEV (non-corrected p-value = 0.003), time coverage (non-corrected p-value = 0.008) and occurrence (non-corrected p-value = 0.048). This suggests that the more emotional difficulties the child has, the more prevalent map B is. We also found negative correlation between children's attention deficits and hyperactivity problems reported in CBCL 1.5-5 and the mean duration of the map C (non-corrected p-value = 0.020), suggesting that the more attention deficits and hyperactivity problems the child has, the shorter the duration of the map C is.

However, no correlation was statistically significant after applying FDR correction. Results of the exploratory correlations are reported in Figure 4."

Q9. The authors state that the microstates template maps obtained in this study were identical to the canonical maps.

a. How was the similarity of the microstates A-E in this study compared to those reported in other studies?

b. Was dissimilarity calculated or was it determined by visual comparison?

c. Jia et al., 2019 and Nagabhushan Kalburgi et al., 2020 have reported unique microstate topographies for microstate D. Were there any group level differences in the template microstate maps in this study?

We thank the reviewer for these relevant questions. In the original and revised manuscripts, the similarity or dissimilarity of the microstates A-E were visually compared to those reported in the literature. Indeed, this was the only possible method of comparison since we do not have access to the raw data of other studies, and hence cannot apply spatial correlation.

Regarding the topography of map D, both Jia et al., 2019 and Nagabhushan Kalburgi et al., 2020 decided a priori to extract four microstate clusters (A, B, C and D) while we chose the number of clusters defined as the best solution for all participants together as recommended by Michel and König, 2018 (Michel and König, 2018). Therefore, we obtain a five-map template as the best solution. Our 4th map was visually similar to the 4th map that our group

has obtained in our previous study (Jan et al., 2019) and we therefore labeled our 4th map as microstate class D and our 5th map as microstate class E, respectively.

Moreover, there are other methodological differences between studies. Both Jia et al., 2019 and Nagabhushan Kalburgi et al., 2020 used T-AAHC clustering method and we used the k-means clustering method with high number of randomizations. Both Jia et al., 2019 and Nagabhushan Kalburgi et al., 2020 have small samples size (15 and 13 children with ASD, respectively) and wide age ranges (5-18 y.o. and 8-14 y.o., respectively) in comparison to our study.

In our study, we applied spatial correlations to compare the 5 maps template extracted from the segmentation of the ASD group with the 5 maps template extracted from the segmentation of the TD group. Pairs of microstate maps were highly similar (Pearson's spatial correlation coefficients for maps A ASD vs TD = 0.992, maps B ASD vs TD = 0.986, maps C ASD vs TD = 0.996, maps D ASD vs TD = 0.958, maps E ASD vs TD = 0.985). Therefore, we used each group's template to compute the temporal parameters of its participants.

In the revised manuscript, the following modifications were made:

Section RESULTS, subsection Microstate analysis " The k-means cluster analysis across all participants identified five dominant maps, which explained 80.1 percent of the total variance (Figure 1).

The topographies of these five cluster maps corresponded to the canonical microstate classes previously reported in the literature and were labeled accordingly (map A, B, C, D and E) (Michel and König, 2018) (Jan et al., 2019). We also applied a k-means cluster analysis across toddlers and preschoolers with ASD and TD peers separately and extract the five cluster maps for both groups. For both groups, the five cluster maps also corresponded to the canonical microstate classes previously reported in the literature and were therefore labeled accordingly (map A, B, C, D and E). The five cluster maps extracted from ASD group explained 80.8 percent of the total variance of EEG recordings of this group, and the five cluster maps extracted from TD group explained 81.9 percent of the total variance of EEG recordings of this second group. The five cluster maps extracted from both groups separately were highly similar (Pearson's spatial correlations coefficients for map A = 0.992, map B = 0.986, map C = 0.996, map D = 0.958, map E = 0.985) and were used for further analysis (Figure 2)."

Q10. Were verbally fluent children with ASD recruited? Please provide subscale scores of the Mullen Scales of Learning and the Vineland Adaptive Behavior Scale-II.

We thank the reviewer for this question regarding the profile of children with ASD we recruited. Indeed, we included heterogenous profile of children with ASD, from very low profile (minimally verbal children, with estimated IQ below 70 and severe autism symptoms) to high functioning children (verbally fluent children, with IQ around 100 and moderate autism symptoms).

All participants were recruited from the Geneva Autism Cohort, a longitudinal cohort of young children (Robain et al., 2020, Franchini et al., 2016). Toddlers and preschoolers were included in the ASD group if the previously established clinical diagnosis was confirmed by exceeding the threshold limit for ASD on ADOS-G (Autism Diagnostic Observation Schedule- Generic) (Lord et al., 2000) or ADOS-2 (Second version) (Lord et al., 2012). The mean Severity Score at ADOS for the toddlers and preschoolers with ASD group was 7.67 ± 1.83 .

Cognitive skills and language are part of the heterogeneity found in autism. Considering the young age of participants, cognitive functioning was assessed using two standardized tools: the Mullen Scales of Early Learning composite score (MSEL) (Lee, 2013), or the Psycho-Educational Profile, third edition (PEP-3) (Schopler et al., 2005). The MSEL is a standardized

assessment used to measure cognitive functioning for children from birth through age 68 months. Information about cognitive functioning includes five subdomains: visual reception, fine and gross motor skills, receptive and expressive language. The total development quotient (total DQ) is referred to as an estimate of overall intelligence. In comparison, the PEP-3 is a standardized tool to assess developmental level of children with developmental disorder, in particular ASD, between 2 and 7.5 years old. It includes the same five domains of the MSEL but combines nonverbal and verbal intelligence scores into the same verbal and preverbal Cognition scale and adds an Imitation skills index. We used either the MSEL or the PEP-3, depending when the children were included in our protocol. We indeed added the MSEL to our standardized battery for cognitive assessment only after 2015. As a result, in the current study, cognitive skills for 55 children with ASD were estimated using the MSEL, and in 10 children with ASD using the Verbal/Preverbal Cognition scale of the PEP-3. Data on cognitive skills were missing for one child.

In total, 37 of 65 children with ASD (56 percent, $F = 5$) obtained a score equal or above 70 (range 71.0 – 132.2). Conversely, 28 of 65 children with ASD (44 percent, $F = 6$) obtained a score below 70, (range 21.9 – 69.7). More precisely, looking through the sub-domains of the MSEL among toddlers and preschoolers with ASD, the mean score for expressive language DQ was 60.37 ± 28.51 (range 9.76 - 124.00) and the mean score for receptive language DQ was 67.01 ± 31.65 (range 9.80 - 126.10).

Looking at the Vineland Adaptive Behavior Scale-II scores, the mean score for the global adaptive functioning level of children with ASD was 80.4 ± 11.5 (range 51.0-110.0). The mean score for the communication subdomain of these children was 80.6 ± 14.3 (range 42.0 - 108.0).

In the revised manuscript, we completed the participants' characteristics in the Table 1. with the participants' scores on the different subdomains of the ADOS, MSEL and CBCL 1.5-5 assessments. As we did not use the Vineland Adaptive Behavior Scale-II when re-doing all the correlations between temporal parameters and clinical measures, we prefer not to add participants' scores in the Table 1. However, if the reviewer suggests that it is still important information, we will be happy to include this information in Table 1.

In the revised manuscript, the following modifications were made:

Section METHOD, subsection Participants “Considering the young age of participants, cognitive functioning was assessed using one of two standardized tools: the Mullen Scales of Early Learning composite score (MSEL) (Lee, 2013) or the Psycho-Educational Profile, third edition (PEP-3) (Schopler et al., 2005). The MSEL is a standardized assessment used to measure cognitive functioning for children from birth through age 68 months. Information about cognitive functioning includes five subdomains: visual reception, fine and gross motor skills, receptive and expressive language. The total development quotient (total DQ) is referred to as an estimate of overall intelligence. In comparison, the PEP-3 is a standardized tool to assess developmental level of children with developmental disorder, in particular ASD, between 2 and 7.5 years old. It includes the same five domains of the MSEL but combines nonverbal and verbal intelligence scores into the same verbal and preverbal cognition scale and adds an imitation skills index. We used either the MSEL or the PEP-3, depending when the children were included in our protocol. The MSEL was added to our standardized battery for cognitive assessment only after 2015. As a result, in the current study, cognitive skills for 55 children with ASD were estimated using the MSEL, and for 10 children with ASD, using the verbal/preverbal cognition scale of the PEP-3 in. Data on cognitive skills were missing for one child with ASD. Finally, to investigate other symptoms that could be present in children with ASD, all parents have completed the Child Behaviors Checklist 1.5-5 years old version (Achenbach & Rescorla, 2000). The CBCL 1.5-5 consists of one hundred questions

exploring five DSM-scales as Attention Deficit/Hyperactivity Problems, Anxiety Problems, Affective Problems, Oppositional Defiant Problems and Pervasive Developmental Problems.

See Table 1 for characteristics of study participants."

Minor Comments:

Q1. The final sample is not gender matched. Please comment.

We thank the reviewer for pointing out this confusion and we apologize for it. Indeed, "gender matched" was not the exact term we should have used and saying that ASD and TD groups do not differ by gender is more accurate. In the revised manuscript, we modified the composition of our groups to include 20 percent females in order to minimize gender bias. Therefore, the two groups did not differ by gender, p -value > 0.9999 .

In the revised manuscript, the following modifications were made:

Section METHOD, subsection Participants "In total, 293 participants were recruited for the experiment. We did not manage to place the cap on 108 participants. The cap was placed on the heads of 185 participants (110 ASD and 75 TD). Out of those, 60 participants (44 ASD and 16 TD) were excluded due to having too many movement-related artifacts, noisy signal, lack of interest or having insufficient number of epochs for subsequent analysis. This was to be expected given the sensory processing issues frequently reported in children with ASD (Kojovic, 2019). In order to minimize gender bias and include an equivalent percentage (20%) of females in both groups, we excluded 12 females from the TD group. As a result, the final sample consisted of 113 participants: 66 toddlers and preschoolers with ASD (11 females; mean age 3.3 years \pm 1.0, range 1.8-5.9) and 47 TD peers (8 females; mean age 3.3 years \pm 1.2, range 1.8-5.8). Groups did not differ by age ($p = 0.8838$) or sex ($p > 0.9999$)."

Q2. Out of the 150 toddlers with successful EEG netting, were there differences in how reliably the net was placed across the participants, i.e. were any participants excluded because the cap was not accurately placed?

The EEG experts always visually inspected the placement of the cap and adjusted it if necessary, prior to starting the recording. Therefore, no participant was excluded from the study for this reason.

In the revised manuscript, the following modifications were made:

Section METHOD, subsection Procedure and task "The EEG expert ensured that the cap was accurately placed and electrodes adjusted to keep impedance values below 50 kOhm prior to starting the recording."

Q3. Parameters studied include the GEV, the mean duration, the frequency of occurrence and the time coverage. If you are studying directed brain network connectivity, why was transition probability/syntax not analyzed?

We thank the reviewer for this relevant question. We agree with the reviewer that it is interesting to study the dynamics syntax of microstate maps while studying directed brain network connectivity. Therefore, in the revised manuscript, we analyzed all transition probabilities from each microstate to any other using Markov chains and we compared them between ASD and TD groups. The results indeed revealed some differences in dynamic syntax between both groups. We found increased transition probabilities from microstate map C to map B in toddlers and preschoolers with ASD compared to their TD peers ($p < 0.001$)- Transition probabilities were also increased in toddlers and preschoolers with ASD

from microstate map E to map D ($p < 0.001$) and to map B ($p = 0.0145$). In parallel, transition probabilities were increased in TD children compared to children with ASD from microstate map C to map D ($p = 0.0245$) and from map E to map C ($p = 0.001$). To facilitate the understanding of the results for the reader, we have added a new figure (Figure 6) summarizing the differences between the ASD and TD groups.

In the revised manuscript, the following modifications were made:

Section METHOD, subsection Statistical analysis "Finally, we analyzed the transition dynamics of EEG microstate maps by computing probabilities of all transitions from each microstate to any other using Markov chains. The observed probabilities were divided by the expected probabilities to account for the variability in occurrence of the states as described in the literature (Bréchet et al., 2019, Tomescu et al., 2018, Lehmann et al., 2005). The transition probabilities between microstate maps were compared between ASD and TD groups applying unpaired t-tests and FDR correction."

Section RESULTS, subsection Transition probabilities "Comparisons of all transition probabilities between each microstate map revealed increased transitions from microstate map C to map B in toddlers and preschoolers with ASD compared to their TD peers ($p < 0.001$) (Figure 6.). Transition probabilities were also increased in toddlers and preschoolers with ASD from microstate map E to map D ($p < 0.001$) and to map B ($p = 0.0145$). In parallel, transition probabilities were increased in TD children compared to children with ASD from microstate map C to map D ($p = 0.0245$) and from map E to map C ($p < 0.001$). However, differences in transition probabilities from microstate map E to B and from microstate map C to D did not survive FDR correction."

Section DISCUSSION "Moreover, transition probabilities normalized by occurrence of microstates revealed increased transitions from microstate map C to B in toddlers and preschoolers with ASD compared to their TD peers, both maps reflecting activity in DMN and VIS, respectively in the literature (Britz et al., 2010). This result supports the hypothesis of hyper-connectivity between VIS and DMN (Gao et al., 2019; Morgan et al., 2019). We also found increased transitions from microstate map E to D in toddlers and preschoolers with ASD compared to their TD peers. Conversely transitions from microstate map E to C w

REVIEWERS' COMMENTS:

Reviewer #1 (Remarks to the Author):

You addressed all of my concerns. I hope this contributes much to the field.

Reviewer #2 (Remarks to the Author):

The authors have made major analytical updates that have addressed all my previous concerns. The manuscript's results and interpretations are of sound nature.